# The influence of dissolved organic matter on the marine production of carbonyl sulfide (OCS) and carbon disulfide (CS$_2$) in the Peruvian upwelling

Sinikka T. Lennartz[1,a], Marc von Hobe[2], Dennis Booge[1], Henry C. Bittig[3], Tim Fischer[1], Rafael Gonçalves-Araujo[4,5], Kerstin B. Ksionzek[4,6], Boris P. Koch[4,6,7], Astrid Bracher[4,8], Rüdiger Röttgers[9], Birgit Quack[1] , Christa A. Marandino[1]

[1] GEOMAR Helmholtz Centre for Ocean Research Kiel, Düsternbrooker Weg 20, 24105 Kiel, Germany
[2] Forschungszentrum Jülich GmbH, Institute of Energy and Climate Research (IEK-7), Wilhelm-Johnen-Strasse, 52425 Jülich, Germany
[3] Leibniz Institute for Baltic Sea Research Warnemünde, Seestraße 15, D-18119 Rostock
[4] Alfred Wegener Institute Helmholtz Centre for Polar and Marine Research, Am Handelshafen 12, 27570 Bremerhaven, Germany
[5] Aarhus University, Department of Bioscience, Frederiksborgvej 399, 4000 Roskilde, Denmark
[6] MARUM Center for Marine Environmental Sciences, Leobener Straße, D-28359 Bremen, Germany.
[7] University of Applied Sciences, An der Karlstadt, 27568 Bremerhaven
[8] Institute of Environmental Physics, University of Bremen, 28334 Bremen, Germany
[9] Helmholtz-Zentrum Geesthacht, 21502 Geesthacht, Germany
[a] now at: Insitute for Biology and Chemistry of the Marine Environment, University of Oldenburg, Oldenburg

*Correspondence to*: Sinikka T. Lennartz (sinikka.lennartz@uni-oldenburg.de)

**Abstract.** Oceanic emissions of the climate relevant trace gases carbonyl sulfide (OCS) and carbon disulfide (CS$_2$) are a major source to their atmospheric budget. Their current and future emission estimates are still uncertain due to incomplete
process understanding and, therefore, inexact quantification across different biogeochemical regimes. Here we present the first concurrent measurements of both gases together with related fractions of the dissolved organic matter (DOM) pool, i.e. solid-phase extractable dissolved organic sulfur (DOS$_{SPE}$, n=24, 0.16±0.04 µmol L$^{-1}$), chromophoric (CDOM, n=76, 0.152±0.03) and fluorescent dissolved organic matter (FDOM, n=35) from the Peruvian upwelling region (Guayaquil, Ecuador to Antofagasta, Chile, October 2015). OCS was measured continuously with an equilibrator connected to an off-
axis integrated cavity output spectrometer at the surface (29.8±19.8 pmol L$^{-1}$) and at four profiles ranging down to 136 m. CS$_2$ was measured at the surface (n=143, 17.8±9.0 pmol L$^{-1}$) and below, ranging down to 1000 m (24 profiles). These observations were used to estimate *in-situ* production rates and identify their drivers. We find different limiting factors of marine photoproduction: while OCS production is limited by the humic-like DOM fraction that can act as a photosensitizer, high CS$_2$ production coincides with high DOS$_{SPE}$ concentration. Quantifying OCS photoproduction using a specific humic-
like FDOM component as proxy, together with an updated parameterization for dark production, improves agreement with observations in a 1D biogeochemical model. Our results will help to better predict oceanic concentrations and emissions of both gases on regional and, potentially, global scales.

## 1 Introduction

Oceanic emissions play a dominant role in the atmospheric budget of the climate relevant trace gases carbonyl sulfide (OCS) and carbon disulfide ($CS_2$) (Chin and Davis, 1993; Kremser et al., 2016). OCS is the most abundant sulfur gas in the atmosphere, and $CS_2$ is its most important precursor. Both gases influence the climate directly (OCS) or indirectly ($CS_2$ by

oxidation to OCS in the atmosphere), as OCS is a major supplier of stratospheric aerosols (Brühl et al., 2012; Crutzen, 1976), which exert a cooling effect on the atmosphere and can foster ozone depletion (Junge et al., 1961; Kremser et al., 2016). Furthermore, OCS has been suggested as a proxy to constrain global terrestrial gross primary production (Campbell et al., 2008; Montzka et al., 2007; Berry et al., 2013). The oceanic emissions of both gases have recently gained interest because they are suggested to account for a missing source of atmospheric OCS (Berry et al., 2013; Kuai et al., 2015;

Glatthor et al., 2015; Launois et al., 2015). *In-situ* measurements of OCS in surface seawater are still limited, but those available suggest that oceanic emissions are too low to fill the proposed gap of 400-600 Gg S yr$^{-1}$ in the atmospheric budget (Lennartz et al., 2017). Still, oceanic emission estimates are associated with high uncertainties (ca. 50%) (Kremser et al., 2016; Whelan et al., 2018). Reducing these uncertainties for present and future emission estimates requires i) increasing the existing field data across various biogeochemical regimes and ii) increasing process understanding and quantification in the

whole water column to facilitate model approaches.

Most of the *in-situ* observations of OCS and $CS_2$ in seawater were reported from the Atlantic Ocean and adjacent seas, and mainly represent surface ocean measurements (see Whelan et al. (2018) for an overview). Here we report the first concurrent measurements in the surface ocean and the water column for both gases from the Peruvian upwelling. This region is one of the most biologically productive regions in the global ocean, due to the upwelling of nutrient rich water. The upwelling

influences the pool of dissolved organic matter (DOM) exposed to sunlight by transporting DOM from the deep ocean to the surface. The DOM pool is relevant in this context, because it contains the precursors and photosensitizers for the photochemical production of OCS and $CS_2$ (Pos et al., 1998; Flöck et al., 1997; Uher and Andreae, 1997). Here we show measurements of chromophoric and fluorescent DOM as well as solid-phase extractable dissolved organic sulfur ($DOS_{SPE}$), in order to further specify drivers of production processes and improve parameterizations of production rates in

biogeochemical models.

Chromophoric DOM (CDOM) is the fraction that absorbs light in the UV and visible range. CDOM contains photosensitizers that absorb light and facilitate photochemical reactions, and can undergo photodegradation itself (Coble, 2007). A part of the CDOM fraction fluoresces (FDOM), i.e. emits absorbed light at a shifted wavelength. Distinct groups of molecules have a specific fluorescence pattern, enabling the molecule classes such as humic substances or proteins (FDOM

components) to be differentiated (Coble, 2007; Murphy et al., 2013). $DOS_{SPE}$ is operationally defined as the dissolved organic sulfur retained by solid-phase extraction (Dittmar et al., 2008). The method favors the retention of polar molecules, which comprise approximately 40 % of the total dissolved organic carbon (DOC) in marine waters (Dittmar et al., 2008). Due to the operational definition, no direct comparison to the CDOM and FDOM pools is possible (Wünsch et al., 2018).

OCS is produced in the surface ocean by interaction of UV radiation with CDOM (Uher and Andreae, 1997), making coastal and shelf regions a hot spot for OCS production (Cutter and Radford-Knoery, 1993). A reaction pathway through an acylradical intermediate in addition to a thiyl (organic RS·) or sulfhydryl (inorganic SH·, from bisulfide) radical pathway has been proposed by Pos et al. (1998) based on incubation experiments. Indeed, the amount of OCS produced has been shown

to depend on CDOM, more specifically the absorption coefficient at 350 nm ($a_{350}$), and a variety of organic sulfur-containing precursors, such as methionine or gluthathione (Zepp and Andreae, 1994; Flöck et al., 1997). $a_{350}$ has been used as a proxy to calculate photochemical production of OCS previously (Preiswerk and Najjar, 2000).  In addition, von Hobe et al. (2003) suggested a relationship between the photoproduction rate constant and $a_{350}$, making the overall photoproduction rate quadratic with respect to $a_{350}$. This dependency is based on the assumption that $a_{350}$ can serve as a proxy for both

photosensitizers and organic sulfur precursors on large spatial scales. Accordingly, a global parameterization for photochemical production was developed based on $a_{350}$, by integrating data from the Atlantic, Pacific and Indian ocean (Lennartz et al., 2017). To improve this parameterization on a regional scale, we tested whether the precursors can be further specified by an easily measurable fraction of the DOM pool (FDOM components, $DOS_{SPE}$), without performing costly and potentially incomplete analysis on the molecular level. In addition, OCS is produced in a light-independent reaction termed

dark production (Flöck and Andreae, 1996; Von Hobe et al., 2001). Two hypotheses exist to date: an abiotic reaction involving thiyl radicals formed by $O_2$ or metal complexes (Pos et al., 1998; Flöck et al., 1997; Flöck and Andreae, 1996), and a coupling to microbial processes during organic matter remineralization (Radford-Knoery and Cutter, 1994). Dark production is parameterized based on temperature and $a_{350}$ derived from field data in the Atlantic Ocean and the Mediterranean Sea (Von Hobe et al., 2001). It is yet unclear whether this parameterization is valid on a global scale.

Furthermore, OCS is degraded by hydrolysis yielding $CO_2$ and hydrogen sulfide ($H_2S$) or bisulfide ($SH^-$), in the following summarized as sulfide. The hydrolysis degradation rate increases strongly with temperature, and has been well quantified by a comprehensive laboratory study over a wide temperature range (Elliott et al., 1989) and by seawater incubation studies (Radford-Knoery and Cutter, 1994). Oceanic OCS concentrations have been modelled using surface box models on regional (von Hobe et al., 2003) and global scales (Lennartz et al., 2017), in the water column (von Hobe et al., 2003) as well as with

a global 3D circulation model (Preiswerk and Najjar, 2000; Launois et al., 2015) based on the same or similar parameterizations as described above. Here we test whether subsurface concentrations can be numerically simulated by coupling the box model to a physical 1D water column host model.

Production and loss processes for $CS_2$ are less well constrained. Photochemical incubation studies indicate that the photoproduction of $CS_2$ has a similar wavelength-dependence (spectrally resolved apparent quantum yield, AQY), but only a

quarter of the magnitude compared to OCS (Xie et al., 1998). It is currently unclear whether *in-situ* photoproduction rates of both gases co-vary on larger spatial scales. A covariation is expected only when identical drivers limit the production for both gases. Evidence for biological production comes from incubation studies (Xie et al., 1999), indicating varying $CS_2$ production for different phytoplankton species. Outgassing to the atmosphere appears to be the most important sink for $CS_2$ in the mixed layer (Kettle, 2000). Although $CS_2$ is hydrolyzed and oxidized by $H_2O_2$, the corresponding lifetimes are too

long to rival emission to the atmosphere at the surface (Elliott, 1990). In addition to the known sinks, namely air-sea exchange, hydrolysis and oxidation, Kettle (2000) proposed a sink with a lifetime on the order of weeks, to match observed concentrations with a surface box model. No underlying mechanism for such a sink is currently known, hampering further model approaches.

5     The goal of this study is to quantify production rates for both gases in the Peruvian upwelling and to further specify their drivers. Surface concentrations and emissions to the atmosphere from the cruise presented here are discussed in Lennartz et al. (2017). Here, we focus on processes in the water column. We use the comprehensive dataset together with simple biogeochemical models to increase the understanding and quantification of the cycling of both gases in the water column and to improve model capability to predict OCS and $CS_2$ seawater concentrations.

## 2 Material and Methods

### 2.1 Study area

The cruise ASTRA-OMZ on RV SONNE started in Guayaquil, Ecuador, on 05.10.2015 and reached Antofagasta on 22.10.2015 (Fig. 1). It covered several regimes from the open ocean to the coastal shelf between 5°N and 17°S. The hydrographic conditions encountered during this cruise have been described elsewhere (Stramma et al., 2016). The area off Peru belongs to one of the four major global eastern boundary upwelling systems (Chavez et al., 2008). A large oxygen minimum zone expands into the Pacific Ocean at depths between 100 and 900 m, resulting from weak ventilation and strong respiration (Karstensen et al., 2008). The cruise covered areas of open ocean with warm sea surface temperatures (SST) between 22-27°C (stations 1-6), and regions with colder SSTs below 20°C closer to the coast (stations 7-18). Upwelling occurred at the southernmost transects indicated by the lowest SSTs (15-18°C) encountered during that cruise (stations 15-18).

### 2.2 Measurement of trace gases

Carbonyl sulfide concentrations were determined with an off-axis integrated cavity output spectrometer (OA-ICOS, Los Gatos Inc., USA) coupled to a Weiss-type equilibrator (Lennartz et al., 2017). The Weiss-type equilibrator was supplied with 2-4 L min$^{-1}$ of seawater from the hydrographic shaft of the ship 5 m below the surface. The sample gas stream from the headspace of the equilibrator was filtered (Pall Acro Filter, 0.2 µm) and dried (Nafion® dryer, Gasmet Perma Pure) before entering the cavity of the OCS analyzer. The outlet of the OCS analyzer was connected to the Weiss-equilibrator, as this recirculation method kept the concentration gradient between the water and gas phases small, enabling rapid equilibration. OCS calibrations using standards from permeation tubes (Fine Metrology, Italy) were performed before and after the cruise, showing good agreement. Details on the OA-ICOS spectrometer can be found in Schrade (2011). The precision of this set-up is 15 ppt for two-minute averages of 1 Hz measurement frequency and was experimentally determined (running a standard for >60 min). The limit of detection is 180 ppt (corresponding to 4 pmol L$^{-1}$ at 20°C), defined by the instrument's internal 1-

second spectra. Additionally, independent samples for comparison measured with GC-MS (Schauffler et al., 1998; de Gouw et al., 2009) reflected <2% difference between the NOAA scale (Montzka et al., 2007) and the perm tube standards. A corrected calibration led to a minor change in absolute concentrations of OCS compared to Lennartz et al. (2017), which was on average +2 pmol L$^{-1}$. Marine boundary layer air was measured every hour for 10 min by pumping air from the ship's deck (ca. 35 m above sea level) through a metal tube (Decabon) with a chemically inert pump (KNF Neuberger). Resulting emissions are reported in (Lennartz et al., 2017).

OCS depth profiles were obtained using a newly developed submersible pumping system. A rotary pump (Lowara, Xylem) connected to a 1" PTFE hose supplied the Weiss-equilibrator with 2-4 L seawater min$^{-1}$. The pump inlet was held at a constant depth for 10-15 min to ensure full equilibration at 4-6 depths during each profile.

CS$_2$ was measured with a purge and trap system attached to a gas chromatograph and mass spectrometer (GC/MS; Agilent 7890A/Agilent 5975C; inert XL MSD with triple axis detector) running in single-ion mode (Lennartz et al., 2017). 50 mL samples were taken in 1 to 3 hour intervals from the same underway system as for continuous OCS measurements. After purging for 15 min with helium (70 mL min$^{-1}$), the gas stream was dried with a Nafion® membrane drier (Gasmet Perma Pure) and trapped with liquid nitrogen for preconcentration. Hot water was used to heat the trap and inject CS$_2$ into the GC/MS. The retention time for CS$_2$ ($\frac{m}{z}$ =76, 78) was 4.9 min. The analyzed data were calibrated daily using gravimetrically prepared liquid CS$_2$ standards in ethylene glycol. During purging, 500 µL gaseous deuterated DMS (d3-DMS) and isoprene (d5-isoprene) were added to each sample as an internal standard to account for possible sensitivity drift between calibrations. The limit of detection was 1 pmol L$^{-1}$. Discrete samples from depth profiles were obtained from the rosette sampler connected to a CTD. Note that OCS and CS$_2$ profiles were not obtained at the same time, but up to seven hours apart. The stations were defined by geographical location and not by a Lagrangian experiment following the same water mass, which explains temperature changes between OCS and CS$_2$ profiles for example at station 2 (see Fig. 2c, S-Table 2).

## 2.3 Chromophoric dissolved organic matter (CDOM)

The spectral absorption coefficient of CDOM ($a_{350}$) was determined for samples collected from the CTD's Niskin bottles or from the underway system, here in a 3-hour interval. The sampled water was filtered through a sample-washed, 0.2 µm membrane (GWSP, Millipore) after pre-filtration through a combusted glass-fiber filter (GFF, Whatman). The optical density of the CDOM in the filtrate was analyzed using a spectrophotometric setup with a liquid waveguide capillary cell (LWCC, WPI; path length: 2.5 m) (Miller et al., 2002). Spectra were recorded for wavelengths between 270 to 700 nm at 2 nm spectral resolution for the sample filtrate and purified water as the reference, with sample and reference at room temperature. The absorption coefficient is determined from the obtained optical density using the Lambert-Beer law and corrected for salinity effect (see (Lefering et al., 2017) for details).

## 2.4 Fluorescent dissolved organic matter (FDOM)

Fluorescent dissolved organic matter (FDOM) was recorded in Excitation-Emission-Matrices (EEMs) with a UV-vis-spectrofluorometer (Hitachi F2700) from filtered seawater samples (0.2 µm, <200 mbar below atmospheric pressure) directly onboard. Excitation wavelengths ranged from 220 nm to 550 nm with a resolution of 10 nm. Emission wavelengths were recorded from 250 nm to 550 nm in 1 nm resolution at a photomultiplier voltage of 400 or 800 V, due to a change of method during the campaign (from 10 Oct 2015 onwards). For both voltages, calibration curves with quinine sulfate (5 to 30 ppb) in sulfuric acid were measured with $R^2$ of 0.9991 and 0.9971, respectively. EEMs were blank subtracted and Raman normalized (Murphy et al., 2013). The values are reported here in quinine sulfate QS units (QSU). A parallel factor analysis (PARAFAC) was performed using the drEEM Toolbox (Murphy et al., 2013; Stedmon and Bro, 2008) to separate the superimposed optical signals of different fluorophores ('components') in the EEMs. FDOM concentrations are reported here in quinine sulfate units (QSU, Murphy et al. (2013)), the conversion factor between QSUs and Raman units is 0.3540 and 0.4256, for each of the QS calibration (i.e. before and after the change in photomultiplier voltage),  respectively. The components were compared to the database OpenFluor (Murphy et al., 2014) to identify similar components from previous studies in other environments.

## 2.5 Solid-phase extractable dissolved organic sulfur (DOS$_{SPE}$)

DOS$_{SPE}$ was sampled from the underway system or from submersible pump profiles directly into glass bottles and filtered through pre-combusted GF/F filters (Whatman, 450°C for >5h) at maximum 200 mbar below atmospheric pressure. 450 mL of each filtered samples were acidified to pH 2 (hydrochloric acid, suprapur, Merck), extracted according to Dittmar et al. (2008) (PPL, 1 g, Mega Bond Elut, Varian) and stored at -20°C until further analysis. For analysis, the PPL-cartridges were eluted with 5 mL of methanol (LiChrosolv, Merck). DOS$_{SPE}$ was quantified with an inductively coupled plasma optical emission spectrometer (ICP-OES, iCAP 7400, Thermo Fisher Scientific). 100 µL of the extract was evaporated with $N_2$ and redissolved in 1 mL nitric acid (1 M, double distilled, Merck). 1 mL of Yttrium (2 µg L$^{-1}$ in the spike solution) was added as internal standard. The sulfur signal was detected at a wavelength of 182.034 nm. Nitric acid (1 M, double distilled, Merck) was used for analysis blank. Calibration standards were prepared from a stock solution (1000 mg L$^{-1}$ sulfur ICP-standard solution, Carl Roth). To assess the accuracy and precision of the method, the SLRS-5 reference standard was analyzed five times during the run. Although sulfur is not certified for SLRS-5, a previous study (Yeghicheyan et al., 2013) reported S concentrations of 2347 – 2428 µg S L$^{-1}$, which is in agreement with our findings. The limit of detection (according to German industry standard DIN 32645) was 1.36 µmol L$^{-1}$ S corresponding to 0.015 µmol L$^{-1}$ DOS$_{SPE}$ in original seawater (average enrichment factor of 89.4).

**2.6 Shortwave radiation in the water column**

Underwater shortwave radiation was assessed through downwelling irradiance profiles obtained with the hyperspectral radiometer RAMSES ACC-VIS (TriOS GmbH, Germany). This instrument covers a wavelength range of 318 nm to 950 nm with an optical resolution of 3.3 nm and a spectral accuracy of 0.3 nm. Measurements were collected with sensor-specific automatically adjusted integration times (between 4 ms and 8 s). Radiometric profiles were collected down to the maximum where light could be recorded prior or after CDOM/FDOM sampling except at Station 7 where sampling took place at night only. Short wave radiation was approximated at this station with the shortwave radiation profile at station 6, which had similar properties in chlorophyll *a* distribution in the water column.

Following the NASA protocols (Mueller et al., 2003), all downwelling irradiance profiles were corrected for incident sunlight (e.g. changing due to varying cloud cover) using simultaneously obtained downwelling irradiance at the respective wavelength, measured above the surface with another hyperspectral RAMSES irradiance sensor. Finally, these data were interpolated on discrete intervals of 1 m.

As surface waves strongly affect measurements in the upper few meters, deeper measurements that are more reliable can be further extrapolated to the sea surface. Each profile was checked and an appropriate depth interval was defined (ranging from 4-25 for Station 2 and 2-25 m for the other three stations) to calculate the vertical attenuation coefficients for downwelling irradiance, [i.e. $K_d(\lambda,z')$] for the upper surface layer. With $K_d(\lambda,z')$ the subsurface irradiance $E_d^-(\lambda, 0\ m)$ were extrapolated from the profiles of $E_d(\lambda,z)$ within the respective depth interval. Finally, short wave radiation rad(z) and photosynthetic active radiation PAR(z) was calculated as the integral over $E_d^-(\lambda, z)$ for λ = 318 to 398 nm and for λ = 400 to 700 nm, respectively, for the depths above the lower limit of the respective depth interval and the originally measured $E_d^-(\lambda, z)$ for the depths below. Finally the euphotic depth $Z_{eu}$ at each station was calculated from the *in-situ* PAR profiles as the 1% light depth where PAR(z) 0.01 of PAR(z=0m).

**2.7 Determination of gas diffusivity with microstructure profiles**

Diapycnal diffusive gas fluxes, i.e. fluxes of dissolved gas compounds caused by turbulent mixing in direction perpendicular to the stratification, were calculated for the four stations 2, 5, 7 and 18. The diapycnal diffusive flux of a compound, $\varphi_{dia}$ [pmol m$^{-2}$ s$^{-1}$], is estimated as

$$\Phi_{dia} \approx \rho \cdot K_\rho \cdot \frac{\partial c}{\partial z} \tag{1}$$

where $\frac{\partial c}{\partial z}$ [pmol kg$^{-1}$ m$^{-1}$] is the vertical gradient of gas concentration across a layer of ideally constant stratification and constant diffusivity, $K_\rho$ [m$^2$ s$^{-1}$] is the diapycnal turbulent diffusivity, and ρ [kg m$^{-3}$] is the water density. Fluxes can be estimated for depth ranges that are limited above and below by concentration measurements, and that do not vary systematically in stratification and turbulent mixing within. Particular focus is on fluxes to/from the mixed layer (ML), which however cause particular issues because of the sudden changes in stratification and mixing intensity at the mixed layer depth (MLD). That is why we approximate ML fluxes by fluxes through a transition zone at 5 to 15 m below the MLD,

following Hummels et al. (2013) Hummels et al. (2013), because stratification there is typically strong and relatively constant. MLD was defined here as the depth where the density has increased by an amount equivalent to a 0.5 K temperature decrease compared to the surface (Schlundt et al., 2014). The diapycnal turbulent diffusivity $K_\rho$ was estimated from the average dissipation rate of turbulent kinetic energy, which in turn was estimated from profiles of velocity microstructure. Details on the methodology to estimate diapycnal fluxes of dissolved substances from microstructure measurements and concentration profiles can be found in Fischer et al. (2013) and Schlundt et al. (2014). The microstructure profiles were obtained with a tethered profiler (type MSS 90D of Sea & Sun Technology).

The depths where fluxes could be estimated were then used as upper and lower bounds of budget volumes. The difference of the diapycnal fluxes in and out of each volume determines convergence or divergence of the diapycnal flux. If other transport processes are negligible and if steady state is assumed, sources/sinks to compensate for the flux divergence/convergence can be determined.

Uncertainties of fluxes have been calculated by error propagation from measurement uncertainties of the gas concentrations and of the average $K_\rho$ values. There are additional uncertainties not quantified, e.g. from the approximation of the average gas gradient, or from the neglect of other gas transport processes than diapycnal mixing. It should be noted that the diffusivity profile only represents current conditions during profiling, and can change on a daily basis due to varying stratification, surface winds etc.

**2.8 Determination of OCS dark production rates**

Dark production rates were determined from hourly averaged measured seawater concentrations shortly before sunrise (i.e. ca. 12-14 hours after concentration maximum of the previous day) or at depths below the euphotic zone. Concentration data from this study and a previous study from the Indian Ocean (Lennartz et al., 2017) were used to calculate dark production rates. The determination of dark production rates relies on the principle that in the absence of light, an equilibrium between dark production and loss by hydrolysis results in stable concentrations (Von Hobe et al., 2001). To ensure approximately steady state conditions, we averaged the concentrations one hour before sunrise and compared to the average of the previous hour. We only considered instances when the concentration before sunrise deviated less than 1 pmol $L^{-1}$ from the previous hour for further calculation. These conditions were met at the beginning of the cruise (7 Oct to 12 Oct), when water temperatures ranged between 21-26°C and corresponding e-folding lifetimes of OCS due to hydrolysis 6 (7 Oct) -12 h (12 Oct). In steady state (early morning or below euphotic zone), dark production $P_D$ [pmol $L^{-1}$ $s^{-1}$] equals loss by hydrolysis $L_H$ [pmol $L^{-1}$ $s^{-1}$], the latter being the product of the steady-state concentration [OCS][pmol $L^{-1}$] and the rate constant $k_h$ [$s^{-1}$] according to eq. 2:

$$P_D = L_H = [OCS] \cdot k_h \qquad (2)$$

The rate constant for hydrolysis, $k_h$ [s$^{-1}$], was calculated according to Elliott et al. (1989), eq. 3 and 4:

$$k_h = e^{(24.3 - \frac{10450}{T})} + e^{(22.8 - \frac{6040}{T})} \cdot \frac{K_w}{a[H^+]} \qquad (3)$$

$$-log_{10}K_w = \frac{3046.7}{T} + 3.7685 + 0.0035486 \cdot \sqrt{S} \qquad (4)$$

with temperature $T$, salinity $S$, $a[H]^+$ the proton activity and $K_w$ the ion product of seawater (Dickinson and Riley, 1979). The temperature dependency of the reaction rate $P_D$ [pmol L$^{-1}$ s$^{-1}$] can be described with an Arrhenius-relationship, resulting in the following equation (eq. 5) in its linearized form:

$$\ln\left(\frac{P_D}{a_{350}}\right) = \frac{a}{T} + b \qquad (5)$$

with $a_{350}$ being the absorption coefficient of CDOM at 350 nm [m$^{-1}$], $T$ the temperature [K] and $a$ and $b$ coefficients describing the temperature dependency of the reaction [-]. The production rate $P_D$ is normalized to $a_{350}$ (von Hobe et al., 2001). The parameters $a$ and $b$ in eq. 5 were derived from $P_D$ (eq. 5) in the Arrhenius-plot to obtain a parameterization for dark production rate in relation to temperature and $a_{350}$.

Biases can potentially be introduced in two ways: 1) neglecting other sinks like air-sea exchange can lead to underestimations of the production rate. With wind speeds of 8 m s$^{-1}$ and MLD on the order of 20-40m, life times due to air-sea exchange are in the order of days to weeks, and hence negligible. 2) Sampling less than two half lives after the maximum concentrations can lead to overestimations of the production rate. For the 11 and 12 October, samples considered for calculation of dark production rates were taken less than two half lives after the concentration maximum of the previous day. Since the concentration changed less than 1 pmol L$^{-1}$ within two hours prior to this sampling, we consider the bias as within the range of the given uncertainty.

## 2.9 Surface box models to estimate photoproduction rate constants

The surface box model for OCS has already been used in Lennartz et al. (2017) to estimate OCS photoproduction rate constants. The model consists of parameterizations for the four processes hydrolysis (Elliott et al., 1989), dark production (Von Hobe et al., 2001), photoproduction (Lennartz et al., 2017) and air-sea exchange (Nightingale et al., 2000). *In-situ* measurements of meteorological, physical and biogeochemical parameters are used as model forcing. Photochemical production was calculated according to eq. 6:

$$\frac{dc_{photo}}{dt} = \int_{MLD}^{0} UV \cdot a_{350} \cdot p \qquad (6)$$

with $\frac{dc_{photo}}{dt}$ being the change in concentration due to photoproduction [pmol L$^{-1}$ s$^{-1}$], UV the irradiance in the UV range [W m$^{-2}$], $a_{350}$ the absorption coefficient of CDOM at 350 nm [m$^{-1}$] and the photoproduction rate constant $p$ [pmol J$^{-1}$]. The model was set up in an inverse mode constrained by time series of OCS measurements $\left(\frac{dc}{dt}\right)$ to optimize the photoproduction rate constant $p$ during each daylight period (13:00 to 23:00 h UTC) with a Levenberg-Marquardt-routine (MatLab version 2015a,

Mathworks, Inc.). The scaling of the rate constant $p$ can be seen as the contribution of the prescursors varying in concentration, as detailed in von Hobe et al. (2003).

An analogous model set-up was developed for $CS_2$, including only the processes of air-sea exchange and photoproduction. The estimated production rate hence compensates the sink of air-sea exchange. Processes without known parameterizations,

such as possible biotic production and a potential (chemical) sink are excluded at this stage (see discussion). More information on the model forcing parameters can be found in the supplementary material (S-Tab 1 and 2).

### 2.10 1D water column modules for OCS and $CS_2$

The Framework for Aqueous Biogeochemical Modelling (FABM) was used to couple the box model to a 1D water column model (Bruggeman and Bolding, 2014) and compare simulated concentrations to observations at stations 2, 5, 7 and 18.

FABM provides the frame for a physical host model and a biogeochemical model, wherein the physical host is responsible for tracer transport and the biogeochemical model provides local source and sink terms. The physical host used here is the General Ocean Turbulence Model (GOTM), which is a 1D water column model simulating hydrodynamic and thermodynamic processes related to vertical mixing (Umlauf and Burchard, 2005). GOTM derives solutions for the transport equations of heat, salt and momentum.

*In-situ* measurements of radiation, temperature, salinity, CDOM and meteorological parameters were used as model forcing to represent conditions under which the concentration profiles were taken. Diurnal radiation cycles and constant meteorological conditions, salinity and water temperature were repeated for 5 days for OCS to obtain stable diurnal concentration cycles and 21 days for $CS_2$ due to its longer lifetime.

The same process parameterizations as for the box models were used as local source and sink terms in the 1D water column

modules for OCS and $CS_2$ in FABM. Photochemical production was calculated in the wavelength-integrated approach (300-400 nm) described above in eq. 6, and in addition in a wavelength-resolved approach. For this purpose, we used *in-situ* measured, wavelength resolved downwelling irradiance profiles together with *in-situ* wavelength-resolved CDOM absorption coefficients to model the photoproduction of both gases in the water column based on previously published apparent quantum yields (AQY) by Weiss et al. (1995) for OCS and by Xie et al. (1998) for $CS_2$. We use the AQY by Weiss

et al., since they were measured at the location closest to our study region (i.e. South Pacific). We assume they reflect the DOM composition in our study region best due to their similarity in $a_{350}$.. We note other observed AQYs (Zepp and Andreae, 1994; Cutter et al., 2004), which vary by up to two orders of magnitude. In addition, the photoproduction rate constant $p$ of OCS in eq. 6 was calculated based on the relationship with FDOM component 2 developed in this study.

In addition, sensitivity tests were performed to further constrain production and consumption processes for $CS_2$. Here we

assessed the sensitivity of the general shape of the profiles and did not focus on exact production rates, since both sink and source processes are too poorly constrained to derive reaction rates from single concentration profiles. Profiles were initialized with the lowest subsurface concentration of the respective measured profile: low enough to be able to assess whether in-situ photoproduction can explain concentration peaks below the mixed layer, but high enough to keep diapycnal

fluxes out of the mixed layer in a reasonable range (in contrast to initializing with 0 pmol L$^{-1}$). The same meteorological conditions that occurred on the day of measurement were repeated for 21 days, i.e. ~2-3 times longer than the lifetime due to air-sea exchange. These sensitivity tests demonstrate 1) the sensitivity of surface $CS_2$ concentrations against diurnal mixed layer variations (simulations X98, X98d, X98s), and 2) the sensitivity of the subsurface $CS_2$ peak against the photoproduction rate constant and wavelength resolution (simulations X98x2, pfit, psfit). Testing the sensitivity against diurnal mixed layer variations is important because surface $CS_2$ concentrations depend on the amount of photochemical production occurring within the mixed layer. Air-sea exchange as the major sink for $CS_2$ within the mixed layer led to a relatively long lifetimes on the order of days to weeks during this cruise, so that the conditions during the days prior to the $CS_2$ profile measurements become important. Simulations with adjusted temperature and salinity profiles with a diurnally varying mixed layer between 10m-25m ('shallow' simulation X98s) and 25-50m ('deep' simulation X98d) were performed. For the second test, demonstrating the sensitivity of the subsurface peak, we chose station 5. This station provides the unique opportunity to assess a profile where the photic zone reaches below the ML, hence photoproduction might occur at depths where the sink of air-sea exchange is absent due to the bottom of the mixed layer acting as a barrier.We used two scenarios to assess the subsurface concentrations with one photoproduction rate constant $p$ across the profile, which is consistent with surface concentrations: 1) a scenario during which the AQY by Xie et al. (1998) is scaled by a factor of 2 to match the surface concentration in a wavelength-resolved approach, and 2) a scenario where $p$ is fitted with a wavelength-integrated approach (eq. 6) with (simulation psfit) and without (simulation pfit) allowing for an additional chemical first-order sink.

An overview of the model experiments is listed in Table 1, more information on the model forcing and set-up can be found in the supplementary material (S-Table 2).

## 3 Results

### 3.1 CDOM, FDOM and $DOS_{SPE}$

DOM showed strong spatial variability in FDOM, but less in the $DOS_{SPE}$ concentration and CDOM absorbance. CDOM, here shown as the absorption coefficient at 350 nm, was on average $a_{350}$=0.15 ±0.03 m$^{-1}$ (coefficient of variation, c.o.v.: 0.2 m$^{-1}$). Highest absorption coefficients were found closest to the continent and in the upwelling-influenced region between 17-20°S (Fig. 2e), as expected in upwelling regions (Nelson and Siegel, 2013). This spatial pattern was consistent with the monthly composite of satellite data (Fig. 1).

Four different components of FDOM, representing groups of similarly fluorescing molecules, were isolated and validated with PARAFAC analysis. Components C1 (average±standard deviation 0.015±0.0119 QSU, c.o.v.: 0.79) and C4 (0.0091±0.0158 QSU, c.o.v.: 1.74) have their fluorescence peak in the UV part of the EEM (see supplements, S-Fig. 1). They resemble the naturally occurring amino acids tryptophane and tyrosine (Coble, 2007). Components C2 (0.0032±0.0027 QSU, c.o.v.: 0.84) and C3 (0.0032±0.0158 QSU, c.o.v.: 0.91) fluoresce in the visible range (VIS-FDOM) of the EEM. Their

fluorescence pattern showed characteristics of humic-like substances, and were abundant especially in the southern part of the cruise, closer to the continent and upwelling region (C2 in Fig. 2f, S-Fig. 1).

Surface $DOS_{SPE}$ only showed minor variations along the cruise track with concentrations of 0.16±0.05 µmol L$^{-1}$ (c.o.v.: 0.31). Highest surface $DOS_{SPE}$ concentrations were found in the 16°S transect connected to an active upwelling cell and in the open ocean part of the cruise (Fig. 2g). $DOS_{SPE}$ concentrations in the water column (not shown) decreased with depth, as also found in the eastern Atlantic Ocean and the Sargasso Sea (Ksionzek et al., 2016). Concentrations decreased from 0.76 (5 m depth) to 0.33 µmol L$^{-1}$ in 100 m at station 7, from 0.62 (25 m ) to 0.49 µmol L$^{-1}$ (125 m) at station 7 and from 0.49 (20m) to 0.28 µmol L$^{-1}$ (115m) at station 18. At station 2, concentrations of 0.89-0.91 µmol L$^{-1}$ were measured at a depth of 50-100m; no surface data is available.

## 3.2 Carbonyl Sulfide (OCS)

### 3.2.1 Horizontal and vertical distribution

OCS surface water concentrations ranged from 6.4 to 144.1 pmol L$^{-1}$ (average 30.5 pmol L$^{-1}$) with strong diurnal cycles as described in Lennartz et al. (2017). Surface concentrations increased towards shelf and coast, and were highest along a shelf transect from 8° to 12° S and connected to a fresh upwelling patch around 16°S (Fig. 2a). Surface concentrations as well as emissions to the atmosphere are described in detail in Lennartz et al. (2017). The concentrations in the water column decreased with depth at stations 2, 7 and 18 to ca. 10 pmol L$^{-1}$ below the euphotic zone with varying gradients. Profiles at stations 7 and 18 ranged down to the oxygen minimum zone, but the concentration profiles did not show any corresponding discontinuity. The shape of the concentration profile for station 5 differed from the other stations: here the profile had a convex shape down to 75 m, and it was the only station where a subsurface concentration peak was recorded at a depth of 136 m (Fig. 3).

### 3.2.2 Dark production

The dark production rates at the surface varied between 0.86 and 1.81 pmol L$^{-1}$ h$^{-1}$ along the northern part of the cruise track, and between 0.16 and 0.81 pmol L$^{-1}$ h$^{-1}$ in the four depth profiles below 50 m. The Arrhenius-type temperature dependency showed significantly increasing dark production rates with increasing temperature (Pearson's test, p=5.66 x10$^{-10}$). Dark production $P_D$ both at the surface and at depth along the cruise track (Fig. 4) is described by the following Arrhenius-equation:

$$P_D = a_{350} \cdot \exp\left(-\frac{15182}{T} + 53.1\right) \qquad (7)$$

The Arrhenius-fit could not be improved using FDOM, $DOS_{SPE}$ or $O_2$ instead of $a_{350}$ (not shown). At station 5, the dark production rates at 50 and 136 m were larger than predicted for the temperature and the $a_{350}$ present (Fig. 4).

The parameterization for dark production previously including only dark production rates from the North Atlantic, Mediterranean and North Sea (Von Hobe et al., 2001) was updated with the data from the Peruvian upwelling and the Indian Ocean, and yields the following semi-empirical equation (eq. 8) (Fig. 4):

$$P_D = a_{350} \cdot \exp\left(-\frac{16692}{T} + 58.5\right) \qquad (8)$$

### 3.2.3 Diapycnal fluxes

The diapycnal fluxes of OCS within the water column were derived from measured concentration and diffusivity profiles. OCS that was produced at the surface was mixed downwards in all four profiles. Diapycnal fluxes out of the mixed layer were always two or three orders of magnitude smaller than emissions to the atmosphere at stations 2, 5 and 7 with diapycnal fluxes of $8.2 \times 10^{-4}$, $2.4 \times 10^{-4}$ and $3.8 \times 10^{-3}$ pmol s$^{-1}$ m$^{-2}$. An exception is station 18, where diapycnal fluxes (0.48 pmol s$^{-1}$ m$^{-2}$) were almost half of the air-sea flux (-1.0 pmol s$^{-1}$ m$^{-2}$).

### 3.2.4 Photoproduction

The photoproduction rate constants according to equation (6) were previously derived from a surface box model and have already been discussed in Lennartz et al. (2017). For days with concurrent measurements of FDOM (7, 8, 9, 10, 13, 16 October 2015), the correlation between photoproduction rate constant and humic-like FDOM C2 was significant (Pearson's test, p=0.014, R²=0.81, Fig. 5a). Measurements of FDOM (and $a_{350}$) during the period used for optimization of the photoproduction rate constant *p* (i.e. daylight period) were averaged for this correlation. The relationship was quantified by the following empirical equation  (9):

$$p = 85.8 \cdot [\text{FDOM C2}] + 828.76 \quad (9)$$

with the photoproduction rate constant *p* [pmol J$^{-1}$] and the concentration of the FDOM component C2 [QSU]. The correlation with $a_{350}$ only explains a variance of R²=0.01 (n=7, i.e. 7, 8, 9, 10, 12, 13, 16 October 2015). R² increases to 0.3, when the respective days for FDOM C2 correlations are considered (p>0.25). C2 and $a_{350}$ were not significantly correlated during these days (p>0.2, R²=0.36), but showed a similar spatial trend all over the cruise track (Fig. 2). Although our experiment was not strictly Lagrangian, $a_{350}$ only changed <0.05 m$^{-1}$ within each respective fitting period. For FDOM C2, only 1-2 measurements per daylight period were available during the days when photoproduction rate constants were fitted, but variations of only 0.003 QSU per day were encountered during high frequency sampling towards the end of the cruise. This relationship thus carries some uncertainties, and will benefit from additional data from other regions.

OCS concentrations in the water column were simulated with the new module in the model environment of GOTM/FABM. While the AQY of Weiss et al. (1995) yielded surface concentrations of a factor 3-6 too small compared to observations, the L17 simulation overestimated concentrations in all cases up to twofold (Fig. 3). Deviations between simulation and

measurements were reduced by using the updated dark production rate of this study and the linear correlation between FDOM C2 and $p$ shown in Fig. 5a (eq. 9, see section 3.2.2). At station 18, surface concentrations were simulated lower than observed. The shapes of the concentration profiles were well reflected in the simulations except at station 5, where the subsurface concentration peaks at 55 m and 136 m were not adequately reproduced. Despite the different magnitude of the wavelength-resolved (W95) and wavelength-integrated (L17, L19) approaches, the shape of the photoproduction profile in the water column did not show major differences.

## 3.3 Carbon Disulfide (CS$_2$)

### 3.3.1 Horizontal and vertical distribution

The surface concentration of CS$_2$ during ASTRA-OMZ was in the lower picomolar range with an average of 17.8 ±8.9 pmol L$^{-1}$ and displayed diurnal cycles only on some (e.g. 7 Oct 2015), but not at the majority of days (Fig. 3). The spatial pattern of sea surface concentrations was opposite to that of OCS, with highest concentrations distant from the shelf and lowest closer to the shore. Highest surface concentrations of CS$_2$ coincided with warm temperatures (Fig. 2b and 2c). Surface temperatures T [°C] and concentrations of CS$_2$ [pmol L$^{-1}$] were binned for daily averages, and yielded the following relationship ($p=0.0026$, $R^2=0.61$) of (eq. 10):

$$[CS_2] = 2.3\,T - 27.2 \tag{10}$$

The concentration profiles of CS$_2$ did not show a steep decrease with depth like OCS, but were more homogeneous (Fig. 6) apart from subsurface peaks below the mixed layer that occurred for example at stations 2, 5 and 18. The concentration in CS$_2$ profiles down to about 200m was distinctly higher in profiles where upwelling did not occur (stations 1 to 13, ~20 pmol L$^{-1}$) compared to stations in the Southern part of the cruise track (stations 15 to 18, ~10 pmol L$^{-1}$). This difference in concentrations throughout the water column reflected the pattern observed at the surface, where high concentrations coincide with high temperatures.

### 3.3.2 Diapycnal fluxes

The diapycnal fluxes of CS$_2$ within the water column revealed highest production at the surface except for station 18. Within the water column, CS$_2$ was redistributed downwards. Small *in-situ* sinks (stations 2, 7, and 18) and *in-situ* sources at different water depths (stations 2 and 18) within the water column were required to maintain convergences/divergences under a steady state assumption. Fluxes out of the ML were 7.6 x 10$^{-4}$, 3.3 x 10$^{-4}$, 1.9 x10$^{-3}$ and 0.98 pmol s$^{-1}$ m$^{-2}$ at stations 2, 5, 7 and 18 and thus 1-3 orders of magnitude smaller than fluxes to the atmosphere. At station 18, diapycnal fluxes out of the ML and emissions to the atmosphere were at a similar magnitude (0.98 and -1.0 pmol s$^{-1}$ m$^{-2}$ respectively).

### 3.3.3 Photoproduction of $CS_2$

Photoproduction rate constants for $CS_2$ were determined using an inverse set up of the surface box model analogous to OCS, but including only photoproduction and air-sea exchange as source and sink terms. The resulting photoproduction rate constants were between 5 to 70 times smaller than those of OCS. Opposite to OCS, the rate constants did not covary significantly with any FDOM component (p>>0.05). A weak trend was detected for $DOS_{SPE}$ (p=0.08, Spearman's r²=0.44, n=8, Fig. 5), all other tested parameters did not show any correlation (FDOM C1-C4, CDOM).

The shape of the $CS_2$ concentration profiles was modelled for four stations (S-Fig. 2, supplements) with the scenarios described in Table 1. Concentrations in the mixed layer of stations 2,5 and 7 using the wavelength resolved AQY from Xie et al. (1998) yielded concentrations 4-6 times lower than observed (simulation X98).

The influence of mixed layer depth variations was tested in simulations X98d and X98s. Surface concentrations differed from the reference simulation X98 by <2.5 pmol $L^{-1}$ (Fig. 7). The shape of the concentration profile, however, was sensitive to mixed layer variations, as indicated by the sensitivity simulations X98d and X98s. In these artificially created test scenarios, concentrations accumulated below the bottom of the deepest mixed layer during the simulation period.

The subsurface concentration peak was investigated with 1) simulation X98x2 with the wavelength-dependent AQY by Xie et al. (1998) scaled by a factor of 2 so that it matches $CS_2$ concentrations in the mixed layer, and 2) simulation pfit and psfit where a photoproduction rate constant in an integrated wavelength-approach (eq. 6) was fitted to observed profiles (corresponding to an evenly distributed AQY across wavelengths from 300-400 nm). Simulation X98x2 does not reproduce the subsurface peak, whereas simulations 'pfit' and 'psfit' are two possible scenarios to reproduce the observed peak (Fig. 7). Photoproduction rates for these simulations are shown in S-Fig. 3 (see supplementary material).

## 4 Discussion

### 4.1 Carbonyl Sulfide

The four profiles at stations 2, 5, 7 and 18 represent the first observations of OCS profiles in the upwelling area off Peru. They do not indicate any connection to a significant redox-sensitive process, as most profiles show a continuous decreasing shape as expected for photochemically produced compounds with a short lifetime in seawater. The independence from dissolved oxygen concentrations is in line with previous findings (Zepp and Andreae, 1994; Uher and Andreae, 1997). Station 5 was the only profile that differed in shape. This profile was measured in an eddy where downward mixing occurred (Stramma et al., 2016), which may explain the increased concentrations at 55 m. Profiles at station 7 and 18 reached down to the sediment, but did not show increased concentrations towards the bottom. Increased sediment inputs, as e.g. reported from estuarine regions (Zhang et al., 1998), apparently do not play a large role in the studied region, and fluxes to the atmosphere are not affected.

The latter study also raises the question of near surface gradients, suggesting that our shallowest measurement depth of 5 m in both profile and underway sampling might underestimate the flux of OCS. On the other hand, strong near surface stratification acts as a barrier for air-sea exchange (Fischer et al., 2019), and could lead to a bias of the OCS flux if the sampling depth is below the barrier. Since it is difficult to perform underway sampling at shallower depths than a few meters, we cannot fully resolve this issue. However, given the low $a_{350}$ compared to coastal and estuary regions as in Zhang et al. (1998), irradiation likely penetrates deeper into the water column in our study region than in the estuary in their study. Hence, photochemical production likely extended further down into the water column, which reduces the problem of underestimating the flux.

Dark production rates of up to 1.81 pmol $L^{-1}$ $h^{-1}$ in our study were at the upper end of the range of previously reported rates in the open ocean (Von Hobe et al., 2001; Ulshöfer et al., 1996; Flöck and Andreae, 1996; Von Hobe et al., 1999), but similar to those from the Mauritanian upwelling region (Von Hobe et al., 1999). Only incubation experiments in the Sargasso Sea showed higher production rates than reported here, ranging between 4-7 pmol $L^{-1}$ $h^{-1}$ (Cutter et al., 2004).Therein, the authors concluded that particulate organic matter heavily influences dark production. Although no sample-to-sample comparison to particulate organic carbon (POC) is possible for our OCS data, the general range of POC during our cruise was 12.1±6.1 µmol $L^{-1}$ (145.2 µg $L^{-1}$), which is much higher than the POC (ca. 41 µg $L^{-1}$) reported from the Sargasso Sea (Cutter et al., 2004). We thus cannot confirm the influence of POC on dark production in the Peruvian upwelling, and do not find a direct biotic influence.

Our results together with previous studies show that tropical upwelling areas are globally important regions for OCS dark production, likely due to the combination of high $a_{350}$ and moderate temperatures (15-18°C). The temperature dependency of the dark production (eq. 7 and 8) is very similar to the one found by Von Hobe et al. (2001) in the North Atlantic, North Sea and Mediterranean (Fig. 4). The similarity points towards a ubiquitous process across different biogeochemical regimes, as the dependence of the production rate on temperature and $a_{350}$ is very similar for an oligotrophic region like the Sargasso Sea (Von Hobe et al., 2001) or the Indian Ocean during the OASIS cruise (Lennartz et al., 2017) and a nutrient rich and biologically very productive region such as the studied upwelling area. The fit in the Arrhenius-dependency could not be improved by other parameters than $a_{350}$, and showed no influence to dissolved $O_2$. The characteristics that make a molecule part of the CDOM pool, i.e. unsaturated bonds and non-bonding orbitals, also favor radical formation. OCS dark production is thus best described using abiotic parameters such as $a_{350}$ and temperature, than biologically sensitive parameters such as dissolved $O_2$ or apparent oxygen utilisation as a proxy for remineralisation. This independence from biotic parameters supports the radical production pathway. The results are in line with findings by Pos et al. (1998) showing that these molecules can form radicals in the absence of light e.g. mediated by metal complexes, and by Kamyshny et al. (2003) showing a positive correlation of dark production rate and temperature. However, the profile at station 5 provides some evidence that an additional process occurs in the subsurface. The concentration peak was visible in the up- and the downcast, but since we only observed it only once, we cannot conclusively rule out that the OCS peak at 136 m is an artefact. Still,

similar subsurface peaks have been reported from stations in the North Atlantic by Cutter et al. (2004). They concluded that dark production is connected to remineralization.

Diapycnal fluxes at stations 2, 5, 7 and 18 indicate downward mixing from the surface to greater depths in all profiles. However, fluxes were several orders of magnitude smaller than emissions to the atmosphere, except for station 18. There, high diffusivities were observed using the microstructure probe, which most likely result from high internal wave activity as indicated by vertical water displacements of up to 30m during four CTD. . Diapycnal fluxes will change diurnally with the shape of the concentration profile and mixed layer variations, hence, the measurements here only represent a snapshot. Still, the difference in magnitudes between air-sea exchange and diapycnal fluxes seems to be valid at varying times of the days and regions in the studied area. Hence, neglecting diapycnal fluxes when calculating OCS concentrations in mixed layer box models leads only to minor overestimations of the concentrations.

An interesting finding is the significant correlation of the photoproduction rate constant $p$ with FDOM C2 (humic-like FDOM), but not with $DOS_{SPE,}$ given a reported correlation of OCS and DOS in the Sargasso Sea where much higher DOS concentrations of ca. 0.4 µmol S $L^{-1}$ were present (Cutter et al., 2004). It should be noted that the method to extract $DOS_{SPE}$ in our study does not recover all DOS compounds, and we cannot exclude the possibility that this influences the missing correlation between $p$ and DOS. In the studied area, OCS photoproduction is apparently not limited by the bulk organic sulfur, but rather by humic substances. The humic-like FDOM component C2 is an abundant fluorophore in marine (Catalá et al., 2015; Jørgensen et al., 2011), coastal (Cawley et al., 2012) and freshwater (Osburn et al., 2011) environments. This FDOM component seems to be especially abundant in the deep ocean (Catalá et al., 2015), which might be the reason for higher C2 surface concentrations in regions of upwelling, as evident in our study (Fig. 2) and reported by Jørgensen et al. (2011). The significant correlation of $p$ with humic-like fluorophores in our study highlights the importance of upwelling and coastal regions for OCS photoproduction.

A significant correlation (i.e., a limitation) of OCS photoproduction with humic-like substances, but not with bulk $DOS_{SPE}$ can be explained by two scenarios: Under the assumption that only organic sulfur is used to form OCS, the limiting factor is contained in the humic-like C2 fraction of the FDOM pool. The sulfur demand (75.8 pmol $L^{-1}$, the orange area in Fig. 7b) would need to be covered entirely by organic, sulfur-containing precursors. The limiting driver of this process is either organic molecules acting as photosensitizers or a sulfur-containing fraction of the DOM pool that correlates with FDOM C2, but not bulk $DOS_{SPE}$. In that scenario, FDOM C2 can be used as a proxy for the OCS photoproduction rate constant. More data from other regions would help to quantify such a relationship. In a second possible scenario under the assumption that both organic and inorganic sulfur can act as a precursor, the sulfur demand could theoretically be covered by the sulfur generated by hydrolysis of OCS (i.e. 85.8 pmol $L^{-1}$, Fig. 7). In this case, FDOM C2 would only be limiting as long as enough organic or inorganic sulfur is present, for example when temperatures are high enough to recycle sulfur directly from OCS, or when other inorganic sulfur sources are present.

Incubation experiments have shown that inorganic sulfur is a precursor for OCS (Pos et al., 1998). It is not clear whether the mechanism proposed therein occurs under environmental conditions, because sulfide concentrations were higher than in

most marine areas, but also yielded much higher OCS production rates in the magnitude of nM hr$^{-1}$ compared to the magnitude of pM hr$^{-1}$ under natural conditions. Furthermore, the conversion of sulfide to sulfate, rather than to OCS, is thermodynamically favored. Based on our data, we cannot resolve the question about the role of inorganic sulfur in OCS photoproduction, but our results are consistent with the reaction mechanism suggested by Pos et al. (1998). Incubation
experiments at environmentally relevant sulfide concentrations, as well as $p$-DOS relationships across different temperature and DOM regimes will help to resolve this issue.

Our results show that FDOM C2 is a good candidate for a proxy for OCS photoproduction, but its sampling coverage is insufficient for global model approaches at the moment. On global scales, where p varies on a broader range than within the area covered by this study, a$_{350}$ might still be an adequate, but not perfect predictor for this variation (Lennartz et al., 2017).
On local scales, the parameterization for $p$ based on $a_{350}$ can be improved using FDOM C2.

In addition, we used parameterizations from previously reported 0D box models and from this study to assess their applicability to biogeochemical models coupled to a 1D physical host model. It should be noted, however, that the surface data shown here have been used, along with other data, to derive the parameterization for the photoproduction rate constant in Lennartz et al. (2017).

Photoproduction rates based on the wavelength-resolved simulation W95 underestimated observed concentrations in all cases. Other AQYs were not tested, but can be interpreted in a relatively straightforward way, since the AQYs of a given spectral shape is proportional to the OCS production and concentration (in steady state). Higher wavelength-resolved AQY as reported by Zepp and Andreae (1994) from the North Sea and the Golf of Mexico, as well as by Cutter et al. (2004) ranged from twofold to up to two magnitudes higher than the ones reported by Weiss et al. (1995). These differences in
magnitude were attributed to the composition of the DOM pool. To reflect this influence of the DOM composition, Lennartz et al. (2017) parameterized the photoproduction rate constant (corresponding to an integrated AQY) to $a_{350}$, following the suggestion by von Hobe et al. (2003) that a$_{350}$ can be used as a proxy for OCS precursors on larger spatial scales. Using this parameterization for photochemical production in the 1Dwater column model (simulation L19) yielded simulated concentrations closer to, but higher than, observations (Fig. 3). Although the absolute concentrations for the AQY W95 did
not match observations due to the reasons outlined above, the shape of the profile fits observations well. The simulations thus support the experimental findings in most of the previously published AQY work, i.e. the highest OCS yield at UV wavelengths for in-situ conditions.

The simulation using the updated dark production rate and scaling $p$ with FDOM C2 (this study, L19) led to simulated concentrations closest to observations. Remaining deviations between simulated and observed profiles occur e.g. at station 5,
possibly due to the reasons discussed above for dark production rates. At station 18, vertical water displacements of up to 30 m during four subsequent CTD casts were observed, most likely due to internal waves. This displacement could violate assumptions inherent to the 1D approach, i.e. influence of horizontal water transport. In general, our results show that simulating OCS concentrations in the water column is possible by applying surface box model parameterizations as local source and sink terms to a physical host model in the upwelling area off Peru with its specific DOM conditions. The

approach is similar to the 1D model by von Hobe et al. (2003) for the Sargasso Sea, but the updated parameterizations yield a higher agreement in shape and actual concentrations of model simulation and observation.

## 4.2 Carbon Disulfide

The $CS_2$ concentrations measured in this study were higher than those observed during an Atlantic transect (Kettle et al.,
2001, average 10.9 pmol L-1, n=744), in the North Atlantic (13.4 pmol $L^{-1}$) and the Pacific (14.6 pmol $L^{-1}$) (Xie and Moore, 1999), but lower than those reported in a more recent transect through the Atlantic (Lennartz et al., 2017). High concentrations of $CS_2$ coincided with elevated temperatures at the surface in our and in previous studies. The significant relationship between surface temperature and $CS_2$ concentrations corroborates previous findings. Xie et al. (1999) found a positive correlation between $CS_2$ concentration and SST for the Pacific and the North Atlantic with a linear relationship of
$[CS_2] = 0.39t + 7.2$ (t=temperature in °C). Daily averages of our data close to the shelf (n=8, from 12 Oct onwards) between 15 and 20 °C fall within this relationship. However, daily averaged concentrations were higher than predicted according to this relationship further away from the coast at the beginning of our cruise at temperatures between 20 and 30°C (n=4). Overall, we confirm that $CS_2$ concentrations increase with increasing temperatures, but the exact relationship varies spatially. Reasons for this relationship could result from e.g. temperature-driven decay of precursor molecules, but remain speculative.
The results are in line with findings by Gharehveran and Shah (2018), who found increased $CS_2$ formation with increasing temperatures in incubation experiments.

The surface box model to determine photoproduction rate constants of $CS_2$ is set-up as a very simple case, including only the processes of photoproduction and air-sea exchange. The rate constant *p* was only fitted for the increase in concentration during daylight, when photoproduction is expected to be much larger than potential other unknown, continuously acting
sources or sinks. The photoproduction rate constant of $CS_2$ was highest when high $DOS_{SPE}$ was present, indicating that the sulfur source might be limiting for this process. Organic sulfur is required to form $CS_2$ even if one S-atom originates from an inorganic S source (like possibly for OCS). A potential mechanism could include a precursor with an existing C-S double or single bond that reacts with either another organic sulfur radical or sulfide. This mechanism would rationalize the correlation with DOS being present for $CS_2$ and not for OCS. Laboratory studies showed that the organic sulfur compounds cystine,
cysteine and (to a lesser extent) methionine are precursors for $CS_2$ photochemistry (Xie et al., 1998). Such organic sulfur-containing molecules are rare in the marine environment (Ksionzek et al., 2016), which can explain the overall lower photoproduction rate constant of $CS_2$ compared to OCS. We found higher $DOS_{SPE}$ concentrations in the upwelling area off Peru compared to other regions, but similar to $DOS_{SPE}$ concentrations in the Mauretanian upwelling reported by Ksionzek et al. (2016). There, elevated $CS_2$ concentrations were reported as well (Kettle et al., 2001). This spatial pattern suggests that
upwelling regions might be hot spots for $CS_2$ photoproduction. It should be considered, however, that the extraction method used cannot recover all DOS compounds in seawater, so that the correlation between $CS_2$ and $DOS_{SPE}$ may be influenced by the DOM composition.

Our simulation X98 at stations 2, 5, 7 and 18 underestimates mixed layer $CS_2$ concentrations, indicating spatial variations of the AQY, most likely due to changes in the DOM composition, as previously found for other gases (OCS: see above, carbon monoxide (Stubbins et al., 2011), DMS (Galí et al., 2016)). These results corroborate findings by Kettle (2000) and Kettle et al. (2001), who showed that the photoproduction of $CS_2$ was underestimated in some regions by the AQY from Xie et al. (1998). The scaling factor was on the order of 1-10 in Kettle's studies, which is in line with our results (factor 2-4). In future model approaches, a photoproduction rate constant (expressing an integrated AQY) would need to be parameterized, and our results suggest that such parameterizations may rely on DOS or, on a global scale, DOC (since DOS covaries globally with DOC).

More detailed simulations were performed for station 5, because at this station, the photic zone extended below the mixed layer. The wavelength-dependence of the photochemical production is assessed with a 1D modelling approach, where the simulations 'X98x2' and 'pfit'/'psfit' reproduce surface concentrations, but differ in their wavelength-dependence of the photoproduction. In the simulation 'X98x2', the wavelength-dependent AQY was scaled to match surface concentrations, but failed to reproduce the observed subsurface peak at station 5, because photoproduction at wavelengths ~400 nm, that penetrate below the ML was too low. In this scenario, another production process is needed to reproduce the observed profile. Similar conclusions were drawn by Xie et al. (1998). They suggested biological production, as the peaks coincided with the peak of chlorophyll a. However, we did neither find any correlation with chlorophyll *a* nor with marker pigments representing various phytoplankton functional types (data source described in Booge et al. (2018)). A potential other dynamic process, e.g. downward mixing, that influences both gases cannot be ruled out, as concentrations for OCS were also higher than predicted around 50 m.

In our simulation 'pfit', a wavelength-integrated approach was adopted (eq. 6). Photoproduction is calculated with the integrated irradiation (300-400nm) and one rate constant, representing a wavelength-integrated AQY. In this simulation, photoproduction occurring at higher wavelengths, that are penetrating deeper into the water column, is higher compared to the wavelength-resolved simulation (see also S-Fig. 3) and leads to the accumulation of $CS_2$ below the mixed layer.

The accumulation occured because the production is detached from the air-sea exchange sink. In this simulation, a period of 6 days was needed to accumulate enough $CS_2$ below the mixed layer to reproduce observed concentrations. This period highly depended on the actual production at wavelengths around 400 nm and can thus vary. With allowing for an additional sink process below the mixed layer (psfit) corresponding to an additional degree of freedom, observations can also be reproduced. Hence, it is possible to explain observed subsurface peaks by 1) photoproduction alone, if higher production is assumed at wavelength around 400 nm, the peak maximum depending on accumulation time and potential additional sink processes, or 2) via an additional production process only occurring shortly below the ML barrier, such as the biological production suggested by (Xie et al., 1999), or 3) by physical downward transport processes related to mixed layer dynamics (given the long $CS_2$ lifetime, such processes could be either slow but continuous mixing processes or strong one-time events such as storms). Slow sinks below the ML would conserve potential higher concentrations advected from surface waters due to the absence of air-sea exchange in the subsurface. The process leading to the observed profiles thus remains inconclusive.

Our results highlight the importance of Lagrangian experiments following the same water mass for compounds with a lifetime on the order of days. Information on the conditions prior to the profile measurements are needed to conclusively interpret the location and accumulation of subsurface peaks.

$CS_2$ was still detectable below 200 m, in concentrations around 5-10 pmol $L^{-1}$ in shelf regions and around 20 pmol $L^{-1}$ in
open ocean regions (except station 1). This pattern reflects the spatial variation of surface concentrations, which were higher in the open ocean than at the shelf. The vertically relatively uniform concentration profiles suggest low degradation rates, and the travel distance of the water between the stations is too short to explain the concentration difference only by *in-situ* degradation. A Lagrangian approach would be helpful to resolve this issue. Some profiles display small local maxima in the region of the oxycline (not shown), but due to unconstrained subsurface source and sink processes, no conclusion can be
drawn on whether a chemical or a physical process is responsible. The rather homogeneous concentrations below 200 m depth suggest slow *in-situ* degradation rates. As a result, physical processes resulting from currents, eddies or shelf processes might gain a higher importance for the distribution of $CS_2$ in the subsurface compared to the shorter lived gas, OCS. With sinks potentially acting on long timescales, $CS_2$ could possibly be transported from sources located further away, e.g. from contact to the sediment in shelf regions or subducted from the surface. Our results clearly show the limits of interpreting 1D
concentration profiles for long lived compounds, with both subsurface sinks and source unconstrained. Incubation experiments using isotopically labelled $CS_2$ would be helpful to constrain source and sink processes independently.

## 5 Summary and conclusion

We show concurrent measurements of the gases OCS and $CS_2$ together with sulfur-containing and optically active fractions of the DOM pool in the upwelling area off Peru. The results indicate how the quality and composition of DOM influences
the production processes of both gases, with implications for predicting their concentrations on regional and, potentially, global scales.

A parameterization for dark production of OCS is updated, resulting in a slight downward correction of the previously established parameterization. The photoproduction rate constant of OCS co-varies regionally with humic-like FDOM, and. more observations of OCS with humic-like DOM could help to improve parameterizations of OCS photoproduction. The
absence of a correlation of the photoproduction rate constant with bulk $DOS_{SPE}$ is not conclusively answered. Possible scenarios involve either a sulfur-containing precursor in a constant ratio FDOM C2, or surplus of sulfur via allocation from OCS hydrolysis. In both cases, FDOM C2 is a promising proxy on regional scales, and in case of scenario 1 also on global scales. In contrast to OCS, the availability of organic sulfur might be a limiting factor for the photochemical production of $CS_2$.

These different limitations of photochemical production of both gases have implications for the expected spatial pattern of their marine surface concentrations. Both, OCS dark and photochemical production, correlate with optically active parts of the DOM pool, which are abundant at high latitudes, coastal and upwelling regions. Also, OCS is degraded by hydrolysis

most efficiently in warm regions such as the tropics, resulting in longer lifetimes in high latitudes. Highest concentrations are thus expected in coastal regions of high latitudes, which is in line with observations. Increasing $CS_2$ photoproduction with increasing $DOS_{SPE}$ concentrations suggests highest surface concentrations in tropical and subtropical regions, where highest DOC and DOS concentrations are expected. This spatial pattern is in line with the limited measurements available (Kettle et

al., 2001). Regarding the tropical missing source of atmospheric OCS, the spatial pattern of oceanic emissions would then favour oxidation of emitted $CS_2$ to OCS as a potential candidate to fill the gap in the atmospheric budget. Our measurements likely represent $CS_2$ concentrations from the upper end of the range of tropical concentrations, since they were performed in a region with high DOS abundance. As an upper limit, a sulfur flux calculated with average values from this cruise (T=20.2°C, S=35, u10=7.3 m s$^{-1}$, $CS_2$=17.8 pmol L$^{-1}$) assumed for the whole tropical ocean (30°N-30°S, 1.95 x10$^{14}$ m²)

results in an annual emissions of 268 Gg S as OCS. This flux, which represents an additional 140 Gg S to the global sulfur flux of $CS_2$ reported by Lennartz et al. (2017) is still too low to sustain a missing source of additional 400-600 Gg S yr$^{-1}$ (800-1000 Gg S$^{-1}$ yr$^{-1}$ total oceanic OCS emissions).

Overall, we show that processes to model OCS distributions are well known and quantified and that the lifetime is sufficiently short to extend the parameterizations of the box model to a 1D water column model. OCS process understanding

is better than for $CS_2$, for which sufficient process understanding to conclusively model subsurface concentrations is still lacking. Our results emphasize the importance of vertical dynamics for longer lived compounds such as $CS_2$ compared to the short lived OCS.

This study highlights the need for more *in-situ* measurements of OCS and $CS_2$ below the mixed layer in various biogeochemical regimes together with fractions of the DOM pool, to improve the suggested quantitative relationships across

larger DOM variations. Subsurface processes, especially for $CS_2$, remain elusive and require concerted experimental and field studies.

**Author contributions**

S.T.L. and C.A.M. designed the study. Measurements and interpretation for essential parameters was performed by D.B. ($CS_2$), T.F. (microstructure profiles), R.G.-A. (PARAFAC), K.B.K. and B.P.K. ($DOS_{SPE}$), A.B. (radiation), R.R. (CDOM

absorption). S.T.L. performed the simulations with support from H.B.. S.T.L., C.A.M. and M.v.H. synthesised the data. S.T.L. wrote the manuscript with contributions from all coauthors.

**Acknowledgements**

We acknowledge the help of the co-chief scientist, Damian Grundle and the crew and captain of the RV Sonne during ASTRA-OMZ to perform our measurements. We thank M. Lomas for providing data on particulate organic carbon. This

work was supported by the German Federal Ministry of Education and Research through the project ROMIC-THREAT

(BMBF- FK01LG1217A and 01LG1217B), ROMIC- SPITFIRE (BMBF- FKZ: 01LG1205C) and SOPRAN, the DFG grants GR4731/2-1 and MA6297/3-1, as well as a PhD grant within the DFG-Research Centre/Cluster of Excellence "The Ocean in the Earth System". Additional funding for C.A.M. and S.T.L. came from the Helmholtz Young Investigator Group of C.A.M., TRASE-EC (VH-NG-819), from the Helmholtz Association through the President's Initiative and Networking Fund and

the GEOMAR Helmholtz-Zentrum für Ozeanforschung Kiel. This work was co-funded by a PhD Miniproposal granted to S.T.L. from the Integrated School of Ocean Sciences (ISOS) of the Cluster of Excellence "The Future Ocean" at Kiel University, Germany, as well as GEOMAR Seedfunding. We thank NASA Goddard Space Flight Center, Ocean Ecology Laboratory, Ocean Biology Processing Group for providing access to satellite data of CDOM from Aqua MODIS. We thank C. Schlundt for her help in analyzing $CS_2$ samples and I. Stimac for her help in analyzing $DOS_{SPE}$ samples. R.G.-A. was

supported by a PhD fellowship from the Coordination for the Improvement of Higher Level Personnel (CAPES-Brazil, Grant 12362/12-3) in collaboration with the German Academic Exchange Service (DAAD).

The authors declare no conflict of interests.

Data/code is available from the corresponding author upon request.

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

**Figures**

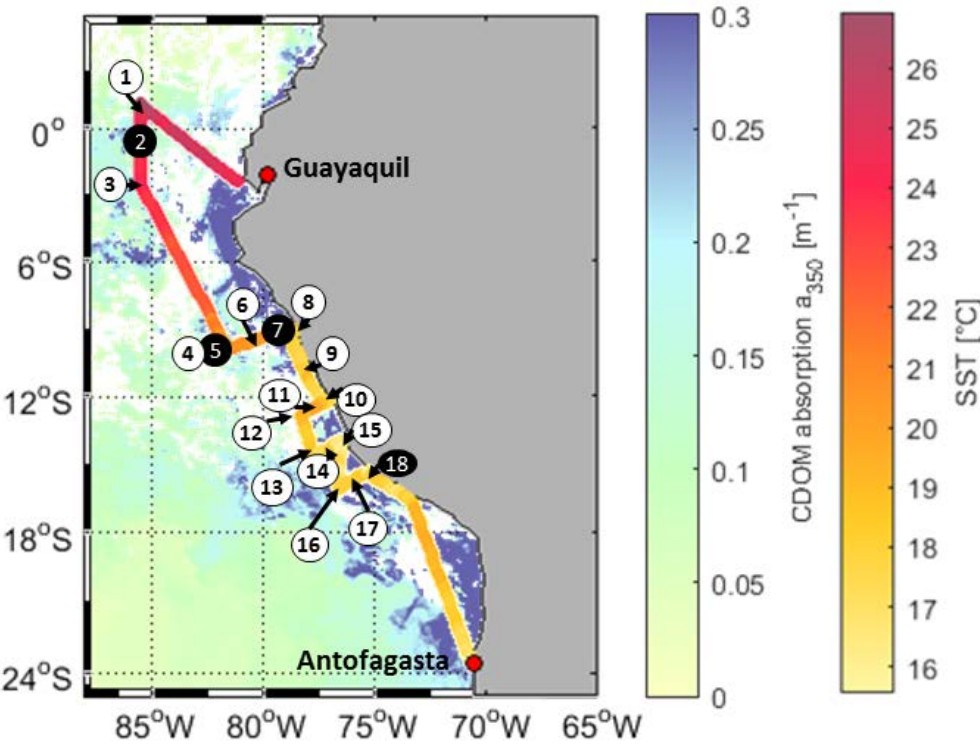

**Figure 1: Cruise track of ASTRA-OMZ with stations 1-18 (in black circles: stations where OCS profiles were taken). The cruise track shows sea surface temperature (SST) measured onboard. For visualization only, the background is Aqua MODIS satellite data for the absorption of CDOM and detritus corrected from 443 nm to 350 nm with the mean slope of our *in-situ* measurements (0.0179, 300-450 nm, Aqua MODIS composite for October 2015). Note: As a monthly composite does not necessarily reflect the exact conditions during the cruise, *in-situ* measurements are illustrated in Fig. 2e. White areas: no satellite data available.**

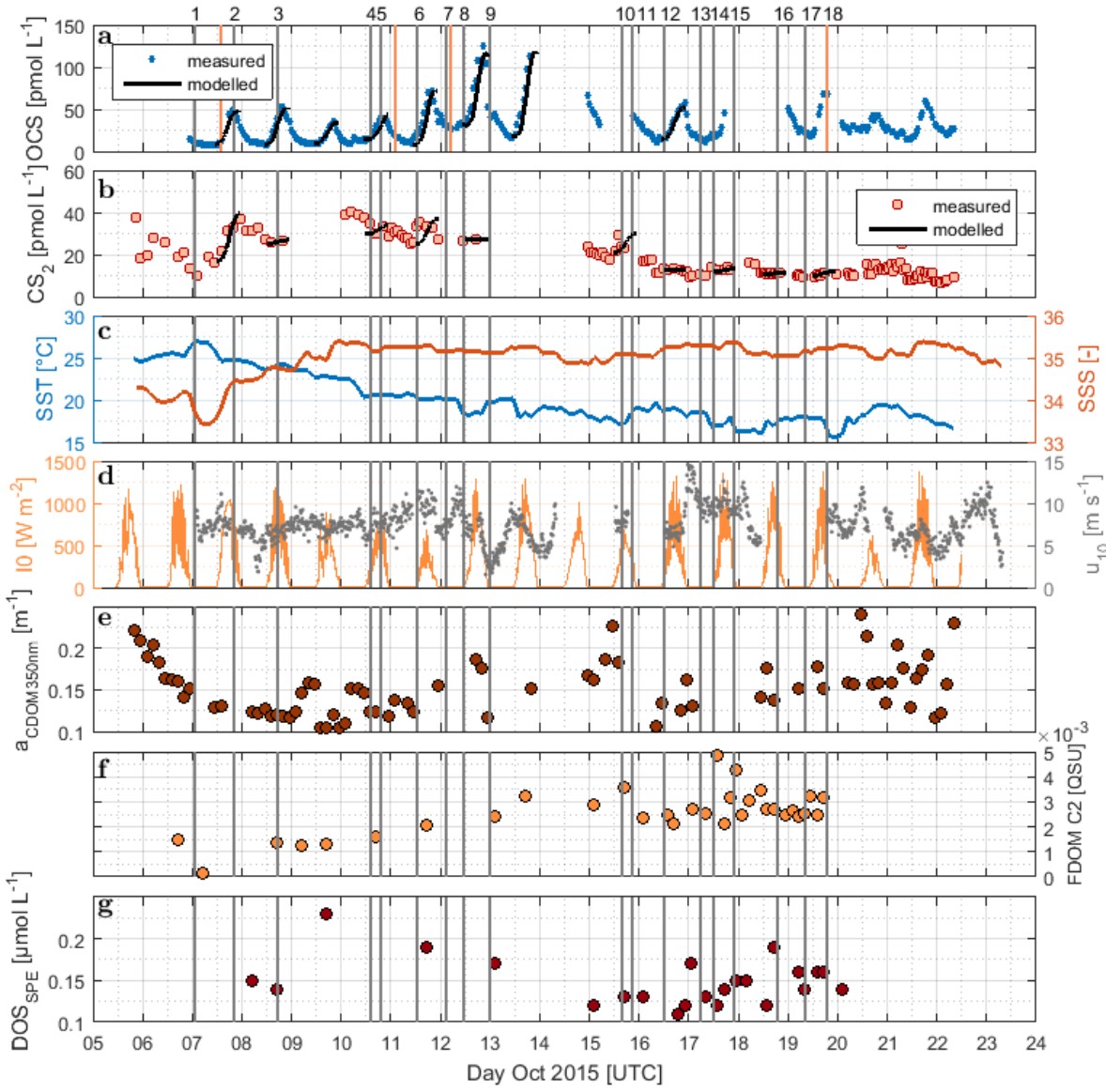

**Figure 2: Time series of a) OCS, b) CS₂, c) SST and SSS, d) I0 and wind speed at 10m, e) absorption coefficient of CDOM at 350 nm, f) humic-like FDOM component 2, and g) DOSSPE sampled from the underway system along the cruise track of ASTRA-OMZ from 5 to 23 October 2018.Vertical lines indicate stations of ASTRA-OMZ for comparison with location (see Fig. 1).**

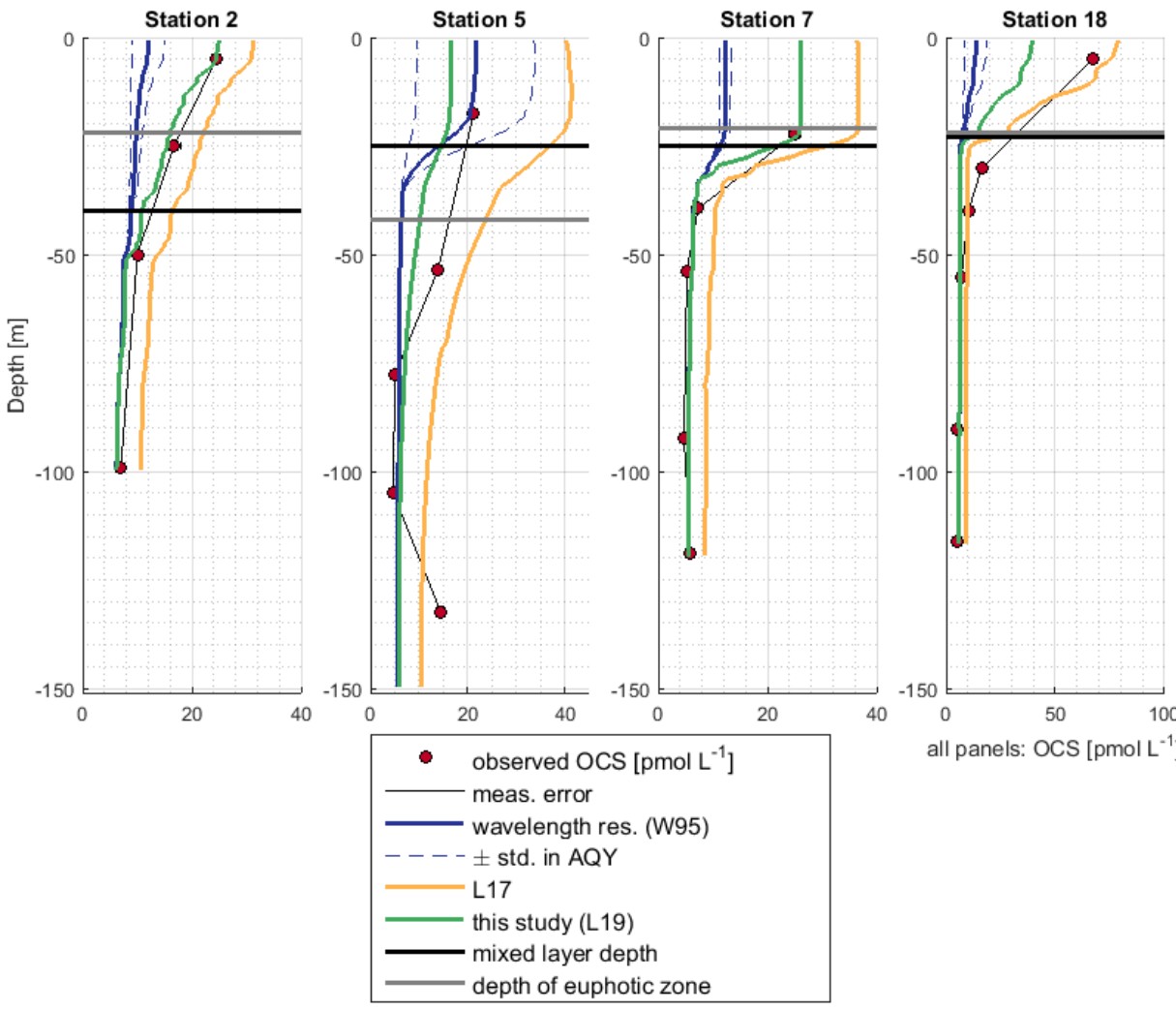

**Figure 3: Profile measurements of OCS concentrations and 1D model results for the OCS model experiments described in Table 1.**

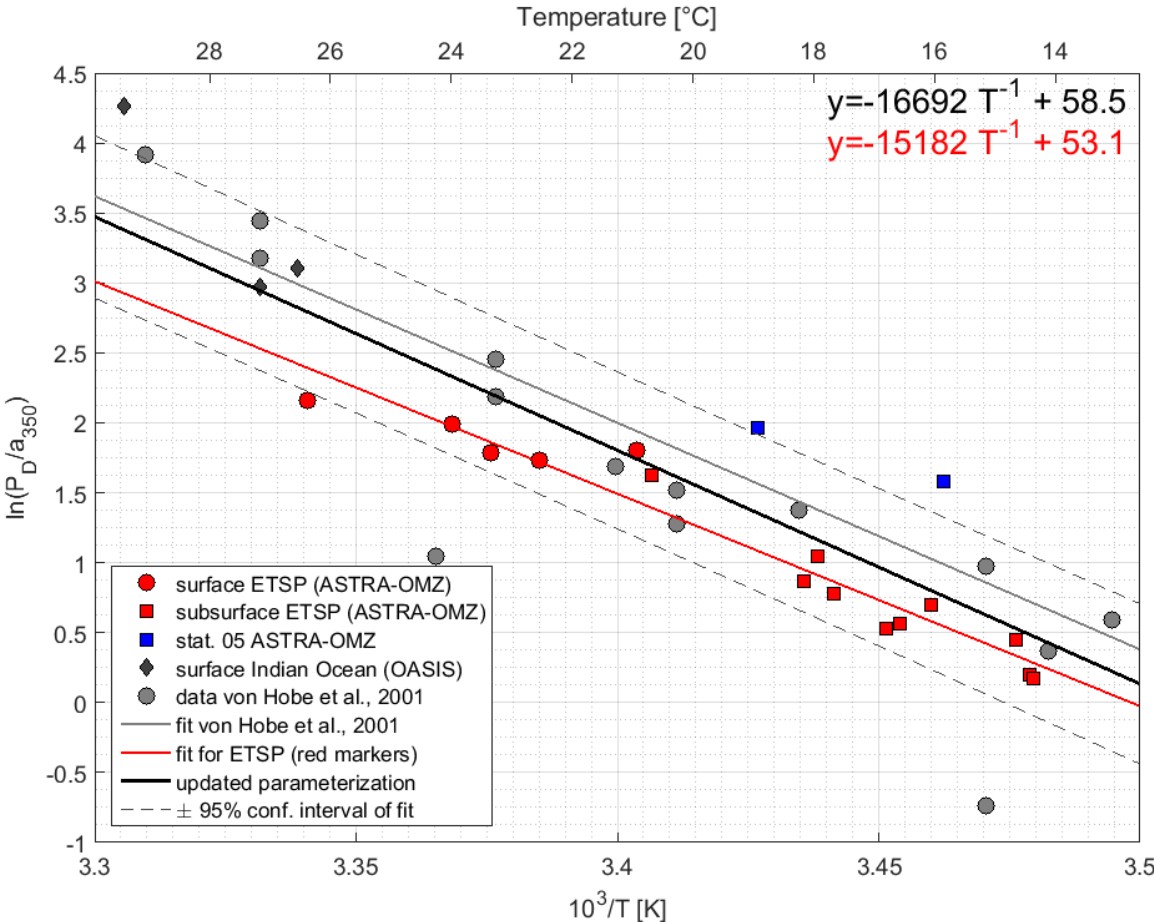

**Figure 4: Arrhenius-plot of dark production rates from ASTRA-OMZ (this study, red and blue markers), data from the Indian Ocean (OASIS cruise, Lennartz et al. (2017)) and previously published rates (von Hobe et al., 2001, grey markers, note that $P_D$ was converted from original units of pmol m$^{-3}$ s$^{-1}$ to pmol L$^{-1}$ h$^{-1}$, for reconversion subtract 1.28). The red linear fit and equation shows the parameterization for ASTRA-OMZ only, whereas the black fit and equation is an updated parameterization including dark production rates from this and previous studies (see Von Hobe et al. (2001)).**

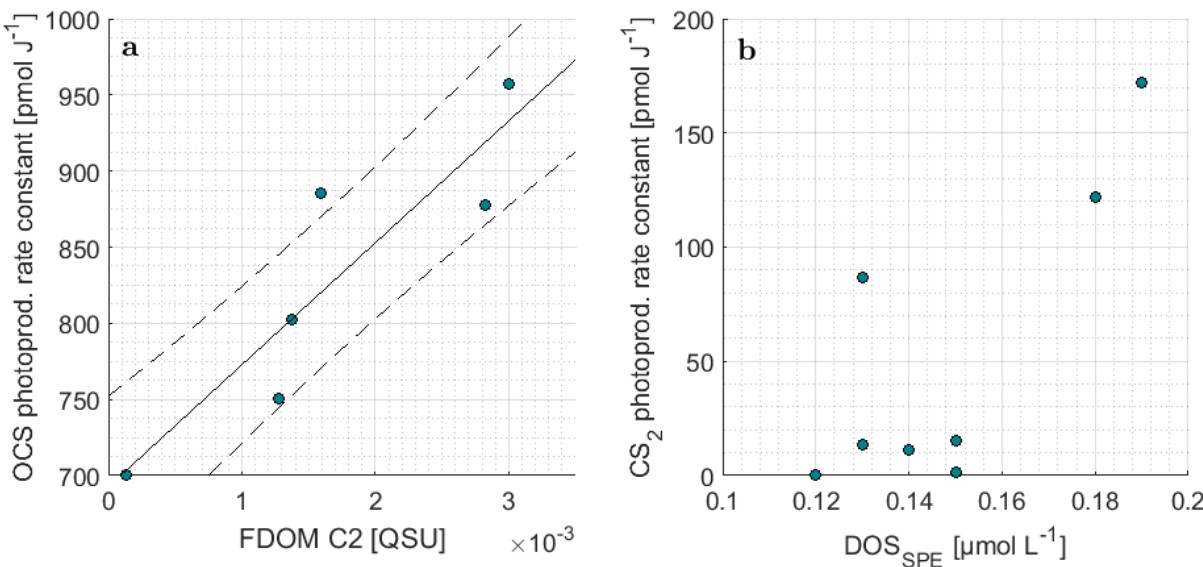

**Figure 5: Correlations of the photoproduction rate constant from inverse surface box modelling for a) OCS and FDOM component C2 and b) $CS_2$ and $DOS_{SPE}$.**

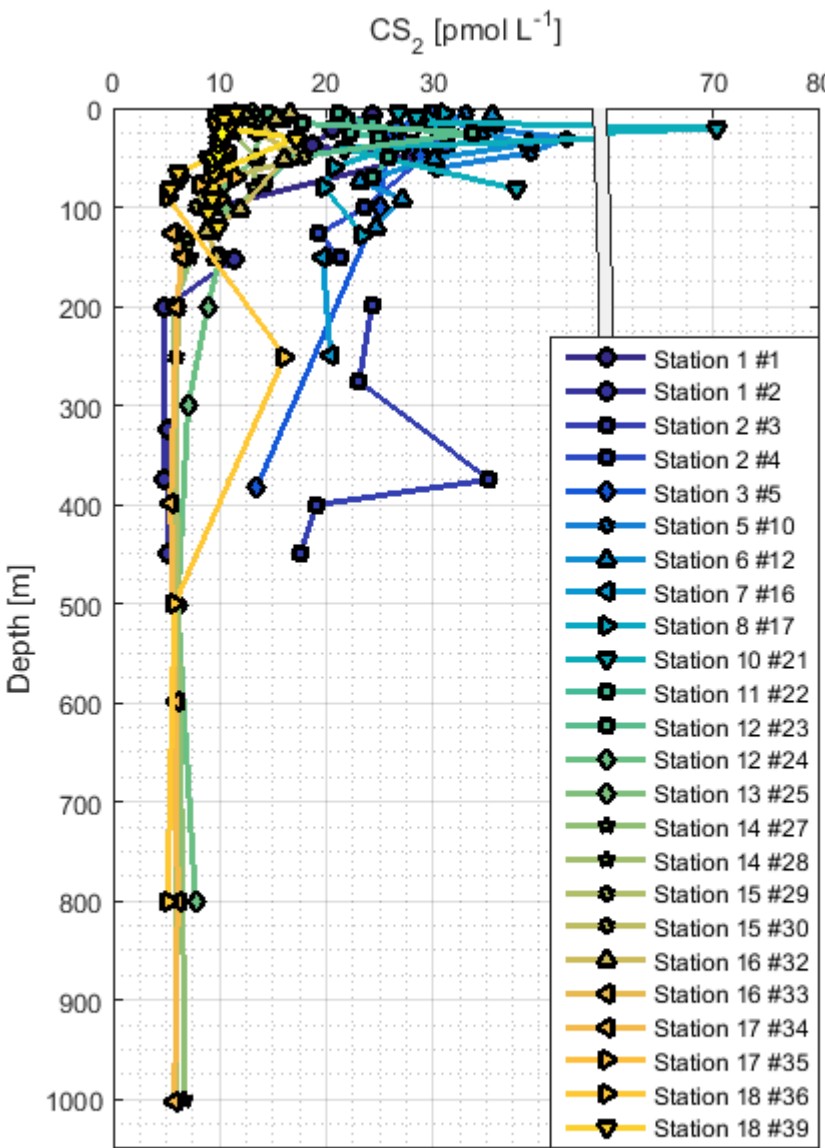

**Fig. 6: Concentration depth profiles for discrete measurements of $CS_2$ for open ocean regions (stations 1-5, blueish colors) and stations closer to the shelf (stations 6-13, green/yellow colors).**

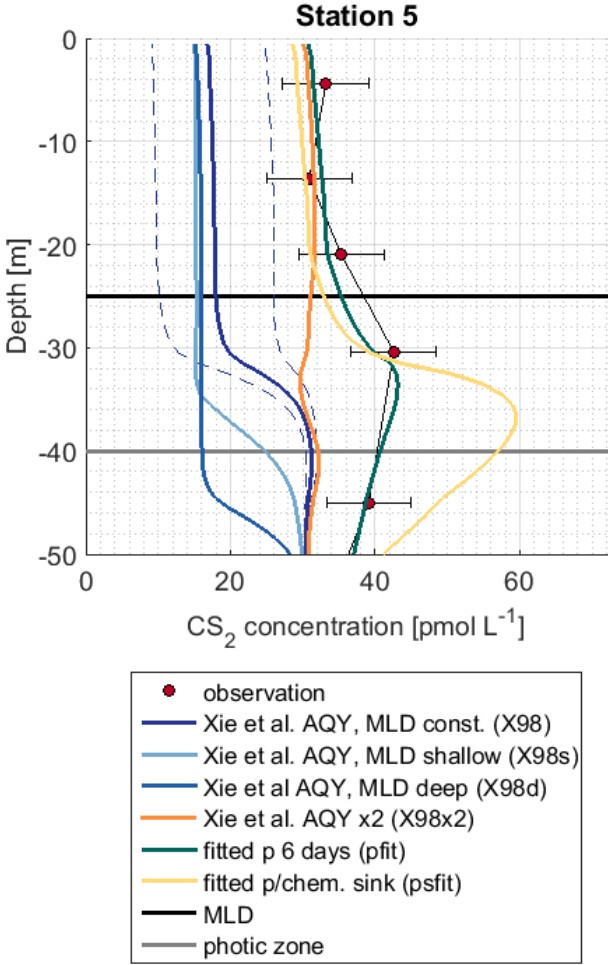

**Figure 7: Observation and model sensitivity simulations at station 5. AQY=apparent quantum yield, MLD=mixed layer depth, chem. Simulation names in brackets refer to Table 1. Dashed lines indicate confidence interval of AQY as reported in Xie et al. (1998).**

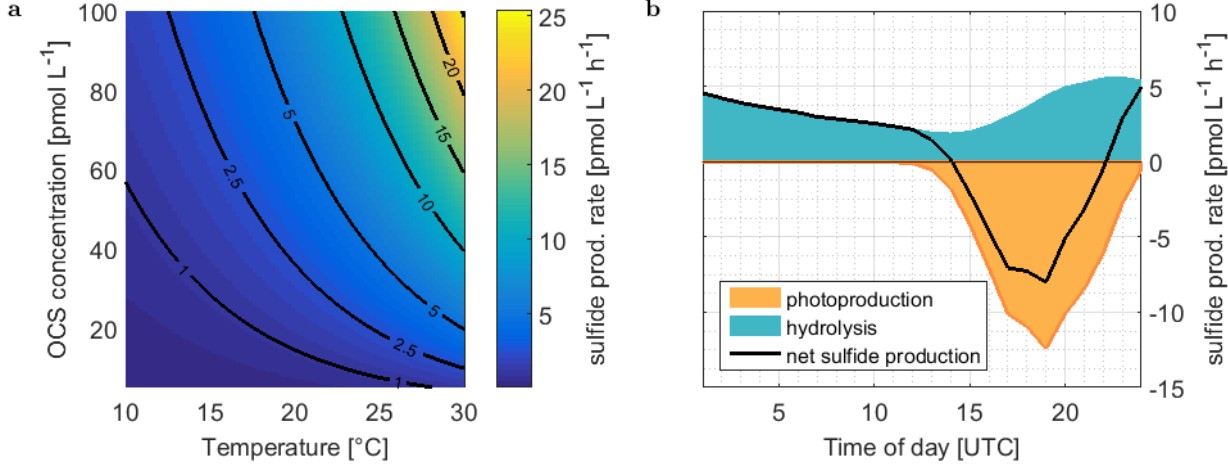

**Figure 8: a) Rate of sulfide production due to OCS hydrolysis as a function of temperature and OCS concentration, calculated with eq. (3)-(4). b) Average consumption of S (organic or inorganic sulfide) by OCS photoproduction and production of sulfide during hydrolysis of ASTRA-OMZ (average 7 October – 14 October).**

**Tables**

**Table. 1: Model experiments with 1D GOTM/FABM Modules for OCS and CS₂. AQY=apparent quantum yield.**

| Carbonyl Sulfide (OCS) | | | | |
|---|---|---|---|---|
| | **Photoproduction** | **Dark prod.** | **Station** | **Description** |
| **W95** | AQY Weiss et al. (1995) | this study | 2,5,7,18 | wavelength resolved photoproduction, mixed layer constant |
| **L17** | Lennartz et al. (2017) | von Hobe et al. (2001) | 2,5,7,18 | wavelength integrated photoproduction, mixed layer constant |
| **L19** | This study ($p$ based on FDOM C2) | this study | 2,5,7,18 | Wavelength integrated photoproduction, mixed layer constant |

| Carbon disulfide (CS₂) | | | |
|---|---|---|---|
| | **Photoproduction** | **Station** | **Description** |
| **X98** | AQY Xie et al. (1998) | 5<br>2,7,18 in supplement | wavelength resolved photoproduction, mixed layer depth constant, no chemical sink |
| **X98d** | AQY Xie et al. (1998) | 5 | wavelength resolved photoproduction, deep diurnal mixed layer variation 25-50m, no chemical sink |
| **X98s** | AQY Xie et al. (1998) | 5 | wavelength resolved photoproduction, shallow diurnal mixed layer variation 10-25m, no chemical sink |
| **X98x2** | AQY Xie et al (1998) x2 | 5 | wavelength resolved photoproduction, mixed layer depth constant, no chemical sink |
| **pfit** | fitted, inverse | 5 | wavelength-integrated (300-400 nm), test for simulation length of subsurface peak, optimized photoproduction rate constant p (eq. 6), no chemical sink |
| **psfit** | fitted, inverse | 5 | wavelength-integrated (300-400 nm), optimized photoproduction rate constant p (eq. 6) and first-order chemical sink. |