# Peer review of "The influence of dissolved organic matter on the marine production of carbonyl sulfide (OCS) and carbon disulfide (CS2) in the Peruvian upwelling"

_Ocean Science, 2019_

## Referee Comment (RC1) · Anonymous Referee #1 · 6 May 2019

Review of manuscript os-2019-1

"The influence of dissolved organic matter on the marine production of carbonyl sulfide (OCS) and carbon disulfide (CS2) in the Eastern Tropical South Pacific" by Lennartz et al.

**1. General comments**

This manuscript presents a comprehensive set of results from a cruise into the Peruvian upwelling area. Observational data are of high quality and unique in that the report OCS and CS2 concentrations together with a wider range of complementary information such as dissolved organic sulfur (DOS), DOM optical characteristics and diapycnal diffusivities. This comprehensive data set is complemented by applications of 1D models of trace gas biogeochemisty and mixed layer dynamics. This approach can allow for near synoptic interpretation although there are of course limitations arising from our incomplete understanding of underlying production mechanisms.

The manuscript is clearly appropriate for publication in Ocean Science, not least due the unique combination of comprehensive data and modeling applications. However, I do have some concerns about presentation, and about some aspects of data interpretation and modeling results.

**Title:** The term 'Eastern Tropical South Pacific' used in the title suggests a study area much larger than that covered in the actual cruise track. I therefore suggest replacing ETSP with 'Peruvian upwelling'.

**OCS photoproduction:** Interestingly the manuscript reports that OCS photoproduction is well correlated with colored dissolved organic matter (CDOM) characteristics such as humic-like fluorescence emission but not with DOS. This leads authors to speculate that the sulfur needed for OCS production may come from its hydrolysis product H2S and not from organic fractions. However, there are several issues the authors neglect to discuss.

Firstly, the absence of a correlation between sulfur from the DOS pool and OCS photo production *per se* does not exclude that DOS fractions are involved in OCS production. Reported DOS concentrations exceed OCS concentrations 1000-fold (see figure 2 in MS). Therefore, only a tiny DOS fraction is required as the source of OCS sulfur. This tiny fraction might not necessarily correlate with bulk DOS. Secondly, although laboratory experiments showed that both H2S and free sulfurbearing amino acids such as cysteine may act as OCS precursors, precursor concentrations in these experiments were unrealistically high (10  $\mu$ M) (Pos et al., 1998). S-OCS may well be derived from DOM itself given that irradiations of both unaugmented seawater and solutions of reconstituted DOM produced OCS (Zepp and Andreae, 1994; Pos et al., 1998).

Thirdly, regarding H2S, available data suggest concentration levels of free sulfide below 100 pM (see e.g. Andreae et al., 1991). Although this may be similar to OCS concentrations, it is unrealistic to assume near complete conversion of

sulfide to OCS. Clearly formation of the thermodynamically stable end product sulfate would be favored here. And finally, although the authors state that sulfurbearing amino acids '*are rare*' (p 17, lines 9-11), they are still likely to occur at levels high enough to sustain picomolar OCS levels.

I recommend that the authors rewrite the corresponding sections in the light of my comments above.

**OCS dark production:** The manuscript presents OCS dark production rates derived using a steady state assumption (methods, p 8). I agree that a steady state assumption may be made for samples *"below the euphotic zone"* or better for samples collected from below the MLD because sea surface OCS shows a pronounced diel cycle. However, I am not entirely convinced that "early morning" samples always reflect steady state. Half lives of OCS with respect to hydrolysis removal range from > 80 hours at 5°C to ~3 hours at 30°C (Elliott et al., 1989). For most of the cruise track half-lives were likely in the order of 10 hours. Given that OCS concentrations peak in the afternoon, and that early morning sampling will occur less than 2 half-lives later, OCS levels are likely biased i.e. more than 25% higher than 'steady state' assuming that hydrolysis is the main removal process. According to equation (2) artificially high OCS levels directly translate into artificially high dark production rates.

Therefore, I would welcome some additional detail on sample selection, and some quantitative considerations of possible bias in section 2.8.

**CS**2 vertical profiles: On p 13 ff the authors present CS2 depth profiles which are then further discussed on p 17 line 22 ff together with modeled profiles. Unfortunately, the full set of profiles is only shown in the supplement, although these are clearly needed to support the results and discussion section. I therefore recommend moving figure S2 into the main paper.

I also have some issues with the modeling results shown both in figures 6 and S2. All simulations in S2 and most in Fig 6 show MLD concentrations below those in underlying waters. To me this seems to indicate that photoproduction in the MLD (the only source term used here) is lower than air sea gas exchange losses. However, if photoproduction is the only source term, why do these profiles indicate increasing concentrations with increasing depth across the thermocline? The modeled profiles in S2 all suggest a CS2 flux from below the TC into the mixed layer, although photoproduction should clearly be constrained to the MLD. Do these profiles show a non-steady state solution? Is it possible that the 1D model was initiated with a homogenous CS2 depth distribution rather than with CS2 free water? I think here the manuscript needs to give a much more detailed explanation of the modeling setup and a more cogent explanation of the resulting profiles.

I am also not entirely convinced of the authors' idea that 'deep' photoproduction below the MLD may have caused the observed subsurface maxima at stations 5 and 18. The statement on p 17 lines 34-5 "*substantial production takes place at higher wavelengths penetrating deeper into the water column*" needs to be put into context: based on apparent quantum yields used (Xie et al., 1998), light levels and KD, what is the depth dependence of CS2 photoproduction? How much of the total occurs below the MLD?

I wonder how this likely very small source term would compare to diapycnal transfer, and I very much doubt that it could sustain the elevated CS2 levels below the MLD.

Of course, the limited understanding of  $CS_2$  cycling hinders the modeling here. However, the treatment of  $CS_2$  depth profiles is rather unbalanced, with some data shown only in the supplement and not fully discussed, while Figure 6 only shows results for one station. The authors might want to consider rebalancing the reporting of their  $CS_2$  results.

**2. Specific and editorial comments**

**Abstract:** The abstract is rather vague, does not give details on time period, sample numbers, and areal extent of the study area, and avoids any quantitative statements. Please rewrite.

**Introduction**

- Page 2 lines 26-7: "[CDOM...] and contains the photosensitizers that absorb light and form radicals for photochemical reactions". This statement should be revised: not all reactive species formed during irradiation of natural water DOM are 'radicals' (e.g. singlet oxygen, triplet states, solvated electrons) nor does CDOM act solely as a photosensitizer given that it undergoes photodegradation itself.
- **Page 2 lines 31-2:** "The method favors the retention of polar molecules, which comprise approximately 40 % of the total dissolved organic carbon (DOC) in marine waters". Please add supporting reference.

**Page 3 lines 5 ff:** "the CDOM absorption 5 coefficient at 350 nm (a350) can serve as a proxy for both photoexcitable carbonyl-groups and organic sulfur precursors making the overall photoproduction rate second-order dependent on a350 (von Hobe et al., 2003)".

(i) a350 may be a proxy for DOM but not specifically for carbonyl groups and DOS. Simple carbonyls show absorption maxima well below 350 nm (e.g. aceton: ~265 nm).

(ii) Relationships between OCS photoproduction and a350 were initially proposed by Uher and Andreae (1997) and subsequently applied to a global model by Preiswerk and Najjar (2000). This should be reflected here.
(iii) It would be worth clarifying that 'second order' here should not be understood in terms of chemical kinetics. Instead, this statement refers to the dual roles of CDOM in light absorption and photochemistry.

Page 3 lines 20 ff: Preiswerk and Najjar (2000) should be added here.

**Page 3 lines 31 ff, CS2 lifetime:** Please clarify if this 'CS2 sink' should be airsea gas exchange or an additional unknown process. Gas exchange would have a matching lifetime in the order of weeks.

**Methods**

- Page 4 lines 23 ff, OCS calibration: Please provide a quantitative comparison of calibrations before and after the cruise. Why is the stated LOD (180 pptv) 12 times higher than the stated precision? What did you mean by "NOAA scale"?
- **Page 4 line 29:** Does "*ca. 35 m*" refer to the height of the inlet above sea level? It would also be interesting to see atmospheric OCS mixing ratios reported somewhere.
- **Page 5 lines 26 ff:** Replace "*spectrophotometer*" with "*spectrofluorometer*". I suspect your statement in line 29 refers to photomultiplier voltage? Please state this clearly.

Explain why you list two conversion factors for conversion between QSU and Raman units.

How did you apply the QSU scale to fluorophores that do not show significant overlap with quinine sulphate emission?

- **Page 6 line 30:** "*downwelling irradiance profiles were corrected for incident sunlight*". This statement is worded incorrectly. I suspect you adopted the profile from station 6 after normalization to sea surface irradiance? Please clarify.
- **Page 7 line 7:** Reference to equation 1 is incorrect. Equation 1 refers to diapycnal flux, not to underwater irradiance. Please insert appropriate equation and update equation numbering.

Page 7 lines 23-25: Please add a supporting reference.

- Page 8 lines 1-2, dark OCS production: Please spell out the units used in equations (2) and (5).
- **Page 10 lines 13-15:** Given MLDs in the order of 30-40 m and piston velocities of maybe 1-2 m d-1, CS2 lifetimes with respect to ventilation losses should be in the order of weeks not days. Please revise your statement.
- Page 10 lines 17-19: Explain why "air-sea gas exchange is absent" at station 5.

**Results**

Page 10 lines 28ff, description of results in figure 2: Your statement that "DOM showed strong spatial variability in FDOM, but less in the DOSSPE concentration and CDOM absorbance" is not supported by data in figure 2. All three variables vary roughly 3-fold during your cruise, including FDOM component 2, particularly if highest & lowest FI were excluded from analysis. Unfortunately, however, mean±stdev are not reported for component 2 (nor for any other components).

You stated that DOS decreased with depth but did not quantify this. Please rewrite this section and include the required statistical and quantitative information.

- **Page 11, OCS distribution:** The authors should clearly state here and in their introduction that OCS sea surface concentrations from this cruise were already published in Lennartz et al (2017).
- Page 12, OCS photoproduction, lines 12 ff: It is interesting that OCS production rates only covary with humic-like fluorescence but not with a350. Please give further details of your analysis: how did you bin C2 and a350 data into days? Did you only use samples obtained between sunrise & sunset? Figure 6 only shows data for 6 days. How many days were available to test for pocs a350 correlations? Given that your experiment was not Lagrangian, how could changes in CDOM characteristics during diurnal cycles have affected your relationships?

I would also be interested to see if  $a_{350}$  and C2 fluorescence were correlated with each other as they usually are (Ferrari and Dowell, 1998). If not, the authors should give possible explanations for this unusual trend.

Page 12, diapycnal fluxes, line 6 ff: Why does the air-sea flux have a different algebraic sign? Please clarify.

Explain what condition might have caused the large diapycnal flux at stn 18.

- **Page 13, CS**2 **distribution:** Reference to Fig 3 in line is incorrect. This should be Fig 2. CS2 depth profiles from the supplement should be moved into this section, because they are needed here to support the discussion.
- **Page 13, diapycnal fluxes:** "Small in-situ sinks (stations 2, 7, and 18) and in-situ sources at different water depths (stations 2 and 18) within the water column were required to maintain convergences/divergences under a steady state assumption."

This statement is unclear. Please explain how your statement relates to your  $CS_2$  depth profiles.

**Page 13, CS**2 **photoprodution:** Did you test for possible correlations between production rates and chl *a* or SST?

**Discussion**

- Page 14, Carbonyl sulfide, lines 11-12: "[profiles] do not indicate any connection to a significant redox-sensitive process". This should be expected given that OCS photoproduction was found to be independent of dissolved oxygen concentration (Zepp and Andreae, 1994; Uher and Andreae, 1997). Please refer to previous work.
- Page 14, line 17: Please remove reference to Ulshöfer et al. (1995) because they did not report dark production rates.
- Page 14, "radical production pathway", lines 25 ff: "A strong similarity across different biogeochemical regimes favors the hypothesis of a radical production pathway, which would be indifferent to the prevailing biological community". This statement is simply wrong and should be removed together with other associated statements.

Available evidence clearly shows that rate constants of reactive radical species with DOM can vary significantly as a function of DOM source / composition (see e.g. McKay et al., 2011). Furthermore, DOM and its colored fraction are indirectly derived from primary production and therefore reflect its distribution pattern (see e.g. Carder et al., 1989), although not necessarily short terms variations.

This unsupported speculation should be replaced by appropriate references to previous work. Kamyshny et al. (2003) for example, proposed a mechanism for dark production of OCS.

That aside, possible differences in dark production rates between stations inside & outside the OMZ more likely reflect DOM compositional differences related to the microbial food web.

- **Page 15, line 5:** "[...] *internal waves led to high diffusivities*". Explain how you arrived at this conclusion. What is the evidence?
- Page 15, correlations, lines 10 ff: see my previous comments on correlation results section 3.2.4.

Lines 18-20: give references supporting your statement regarding covariation of humic like fluorescence and  $a_{350}$ .

- **Page 15, lines 21 ff:** see my previous comments regarding H2S as a possible substrate or OCS production.
- Page 16, lines 7 ff: This discussion of 1D model results refers to 'scaling factors' for AQYs. Note that AQYs cannot be assumed to be 'constant'. There is now clear evidence that AQY can vary with CDOM absorbance, presumably due to changes in DOM composition reflected by optical properties. See for example Gali et al. (2016) and Stubbins et al. (2011). The authors' own pocs-CDOM relationships do reflect this as well.

Establishing AQY-CDOM relationships for OCS would be useful and should be discussed in this section.

- Page 16, CS2 vs SST, lines 30 ff: Report the CS2 vs SST relationship in your data and compare this to Xie et al (1999).
- **Pages 17-18, CS2 photoproduction:** see my previous comments regarding CS2 production and vertical profiles.
- Page 18, lines 7-9: Sustained subsurface maxima cannot be caused by 'downward mixing' because the CS2 gradient would cause transfer into the surface layer. Please remove this statement.

**Editorial:**

The wording could be improved by careful editing.

**References**

- Andreae, T.W., Cutter, G.A., Hussain, N., Radford-Knœry, J., 1991. Hydrogen sulfide and radon in and over the western North Atlantic Ocean. J. Geophys. Res. 96, 18,753-718,760.
- Carder, K.L., Steward, R.G., Harvey, G.R., Ortner, P.B., 1989. Marine humic and fulvic acids: Their effects on remote sensing of ocean chlorophyll. Limnology and Oceanography 34, 68-81.
- Elliott, S., Lu, E., Rowland, F.S., 1989. Rates and mechanisms for the hydrolysis of carbonyl sulfide in natural waters. Environ. Sci. Technol. 23, 458-461.
- Ferrari, G.M., Dowell, M.D., 1998. CDOM absorption characteristics with relation to fluorescence and salinity in coastal areas of the southern Baltic Sea. Estuarine Coastal and Shelf Science 47, 91-105.
- Gali, M., Kieber, D.J., Romera-Castillo, C., Kinsey, J.D., Devred, E., Perez, G.L., Westby, G.R., Marrase, C., Babin, M., Levasseur, M., Duarte, C.M., Agusti, S., Simo, R., 2016. CDOM Sources and Photobleaching Control Quantum Yields for Oceanic DMS Photolysis. Environ Sci Technol 50, 13361-13370.
- Kamyshny, A., Goifman, A., Rizkov, D., Lev, O., 2003. Formation of carbonyl sulfide by the reaction of carbon monoxide and inorganic polysulfides. Environ. Sci. Technol. 37, 1865-1872.
- McKay, G., Dong, M.M., Kleinman, J.L., Mezyk, S.P., Rosario-Ortiz, F.L., 2011. Temperature dependence of the reaction between the hydroxyl radical and organic matter. Environ Sci Technol 45, 6932-6937.
- Pos, W.H., Riemer, D.D., Zika, R.G., 1998. Carbonyl sulfide (OCS) and carbon monoxide (CO) in natural waters: evidence of a coupled production pathway. Mar. Chem. 62, 89-101.
- Preiswerk, D., Najjar, R.G., 2000. A global, open-ocean model of carbonyl sulfide and its air-sea flux. Global Biogeochem. Cycles 14, 585-598.

- Stubbins, A., Law, C.S., Uher, G., Upstill-Goddard, R.C., 2011. Carbon monoxide apparent quantum yields and photoproduction in the Tyne estuary. Biogeosciences 8, 703-713.
- Uher, G., Andreae, M.O., 1997. Photochemical production of carbonyl sulfide in North Sea water: A process study. Limnology and Oceanography 42, 432-442.
- Ulshöfer, V.S., Uher, G., Andreae, M.O., 1995. Evidence for a winter sink of atmospheric carbonyl sulfide in the northeast Atlantic Ocean. Geophysical Research Letters 22, 2601-2604.
- Xie, H., Moore, R.M., Miller, W.L., 1998. Photochemical production of carbon disulphide in seawater. J. Geophys. Res.
- Zepp, R.G., Andreae, M.O., 1994. Factors affecting the photochemical production of carbonyl sulfide in seawater. Geophysical Research Letters 21, 2813-2816.

**End of review**

---

## Referee Comment (RC2) · Gregory Cutter (Referee) · 6 May 2019

Given interest in the global radiation balance in a changing world, this manuscript examines the production of carbonyl sulfide and related gas carbon disulfide in waters off Chile using a combined field and modeling approach. It is a relevant and needed compilation for ocean and atmospheric scientists. One of the issues driving any recent examination of sulfur gas fluxes from the ocean is the apparent imbalance between known/established source fluxes, the atmospheric inventories, and residences times – the known fluxes cannot account for the measured inventories. In this respect, this

paper doesn't solve the problem, but in fact they also missed the papers by Cutter and Radford-Knoery (Mar. Chem., 43: 225-233, 1993) that clearly demonstrate the importance of coastal OCS fluxes, and another by Zhang and Cutter (Mar. Chem., 61:127-142, 1998) that shows coastal sediments are a large source to the water column of OCS via sulfate reduction and enhance the sea-air flux. Another interesting feature in this 1998 paper is that low depth-resolution bottle sampling that have acquired OCS depth profiles to date, and that used here with a pump, may be missing large subsurface maxima in OCS that would then radically change the calculated fluxes. These same features could be present for CS2 as well.

In these authors' computations of photochemical and dark production, it would have been beneficial to examine the carefully measured and calculated rates in the Sargasso Sea (Cutter et al., 2004). In this respect, the Sargasso Sea dark production not only depends on the abundance of particulate organic matter, but also is ca. 3x higher than those reported here. Since they have endeavored to compare their results with others, this bears mentioning. Interestingly, their photochemical model using Weiss et al.'s 1995 Apparent Quantum Yields is 3-6 times lower than required based on field data (p. 12, line 22), but the Cutter et al. (2004) AQY parameters would nicely fit their field results. Thus, expanding their search for parameterizations would have solved some of their problems.

Finally, their speculation on p. 15 that dissolved H2S in the surface ocean could maintain OCS abundances is interesting, but in fact the major pathway for oxic H2S production is phytoplankton emissions coupled to assimilatory sulfate reduction, not OCS hydrolysis, and most of the resulting H2S is complexed with trace metals such as zinc (Walsh et al., Limnol. Oceanogr.,39: 941-948, 1994; Radford-Knoery [NOTE CORRECT SPELLING] and Cutter, 1994; Cutter et al., Deep-Sea Res. II, 46: 991-1010, 1999).

---

## Author Comment (AC2) · 17 Jun 2019

**We thank Dr. Cutter for his review, and reply to the comments below.**

Given interest in the global radiation balance in a changing world, this manuscript examines the production of carbonyl sulfide and related gas carbon disulfide in waters off Chile using a combined field and modeling approach. It is a relevant and needed compilation for ocean and atmospheric scientists. One of the issues driving any recent examination of sulfur gas fluxes from the ocean is the apparent imbalance between known/established source fluxes, the atmospheric inventories, and residences times – the known fluxes cannot account for the measured inventories. In this respect, this Discussion paper doesn't solve the problem, but in fact they also missed the papers by Cutter and Radford-Knoery (Mar. Chem., 43: 225-233, 1993) that clearly demonstrate the importance of coastal OCS fluxes, and another by Zhang and Cutter (Mar. Chem., 61:127- 142, 1998) that shows coastal sediments are a large source to the water column of OCS via sulfate reduction and enhance the sea-air flux. Another interesting feature in this 1998 paper is that low depth-resolution bottle sampling that have acquired OCS depth profiles to date, and that used here with a pump, may be missing large subsurface maxima in OCS that would then radically change the calculated fluxes. These same features could be present for CS2 as well.

**We are aware that such a comprehensive, but locally constrained study cannot solve the problem on global emissions alone, but we believe that increased process understanding and testing of model parameterizations in different biogeochemical regimes helps to improve global modeling approaches and, hence, global emission estimates.**

**Concerning emission estimates: Emission estimates that are derived from modelling approaches (i.e. not simply scaling oceanic OCS measurements to the global ocean area) already account for elevated OCS emissions at the coasts, because they take into account input data with e.g. elevated $a_{350}$ in coastal areas (Lennartz et al., 2017). Coastal emissions, although important, are unlikely to account for the whole missing source of 400-600 Gg S yr$^{-1}$.**

**We thank Dr. Cutter for pointing us to the studies, and we include them in the following.**

**p. 3, l. 1: OCS is produced in the surface ocean by interaction of UV radiation with CDOM (Uher and Andreae, 1997), making coastal and shelf regions a hot spot for OCS production (Cutter and Radford-Knoery, 1993).**

**p. 14, l. 15: Profiles at station 7 and 18 reached down to the sediment, but did not show increased concentrations towards the bottom. Increased sediment inputs, as e.g. reported from estuarine regions (Zhang et al., 1998), apparently do not play a large role in the studied region, and fluxes to the atmosphere are not affected.**

**P 14, l. 15ff: The latter study also raises the question of near surface gradients, suggesting that our shallowest measurement depth of 5 m in both profile and underway sampling might underestimate the flux of OCS. On the other hand, strong near surface stratification acts as a barrier for air-sea exchange (Fischer et al., 2019) and could lead to a bias of the OCS flux, if the**

**sampling depth is below the barrier. Since it is difficult to perform underway sampling at shallower depths than a few meters, we cannot fully resolve this issue. However, given the low $a_{350}$ compared to coastal and estuary regions as in Zhang et al. (1998), irradiation likely penetrates deeper into the water column in our study region than in the estuary in their study. Hence, photochemical production likely extended further down into the water column, which reduces the problem of underestimating the flux.**

In these authors' computations of photochemical and dark production, it would have been beneficial to examine the carefully measured and calculated rates in the Sargasso Sea (Cutter et al., 2004). In this respect, the Sargasso Sea dark production not only depends on the abundance of particulate organic matter, but also is ca. 3x higher than those reported here. Since they have endeavored to compare their results with others, this bears mentioning.

**We added a paragraph to discuss these results:**

**p. 14, l. 18: Only incubation experiments in the Sargasso Sea showed higher production rates than reported here, ranging between 4-7 pmol L$^{-1}$ h$^{-1}$ (Cutter et al., 2004).Therein, the authors concluded that particulate organic matter heavily influences dark production. Although no sample-to-sample comparison to particulate organic carbon (POC) is possible for our OCS data, the general range of POC during our cruise was 12.1±6.1 µmol L$^{-1}$ (145.2 µg L$^{-1}$), which is much higher than the POC (ca. 41 µg L$^{-1}$) reported from the Sargasso Sea (Cutter et al., 2004). We thus cannot confirm the influence of POC on dark production in the Peruvian upwelling, and do not find a direct biotic influence.**

Interestingly, their photochemical model using Weiss et al.'s 1995 Apparent Quantum Yields is 3-6 times lower than required based on field data (p. 12, line 22), but the Cutter et al. (2004) AQY parameters would nicely fit their field results. Thus, expanding their search for parameterizations would have solved some of their problems.

**We added:**

**Methods:**

**p. 10, l. 4: We use the AQY by Weiss et al., since they were measured at the location closest to our study region (i.e. South Pacific). We assume they reflect the DOM composition in our study region best due to their similarity in $a_{350}$. We note other observed AQYs (Zepp and Andreae, 1994; Cutter et al., 2004), which vary by up to two orders of magnitude.**

**Discussion:**

**p. 16, l. 11: Photoproduction rates based on the wavelength-resolved simulation W95 underestimated observed concentrations in all cases. Other AQYs were not tested, but can be interpreted in a relatively straightforward way, since the AQYs of a given spectral shape is proportional to the OCS production and concentration (in steady state). Higher wavelength-resolved AQY as reported by Zepp and Andreae (1994) from the North Sea and the Golf of Mexico, as well as by Cutter et al. (2004) ranged from twofold to up to two magnitudes higher than the ones reported by Weiss et al. (1995). These differences in magnitude were attributed to the composition of the DOM pool. To reflect this influence of the DOM composition, Lennartz et al. (2017) parameterized the photoproduction rate constant (corresponding to an integrated AQY) to**

$a_{350}$, following the suggestion by von Hobe et al. (2003) that a$_{350}$ can be used as a proxy for OCS precursors on larger spatial scales. Using this parameterization for photochemical production in the 1D water column model (simulation L19) yielded simulated concentrations closer to, but higher than, observations (Fig. 3). Although the absolute concentrations for the AQY W95 did not match observations due to the reasons outlined above, the shape of the profile fits observations well. The simulations thus support the experimental findings in most of the previously published AQY work, i.e. the highest OCS yield at UV wavelengths for in-situ conditions.

Finally, their speculation on p. 15 that dissolved H2S in the surface ocean could maintain OCS abundances is interesting, but in fact the major pathway for oxic H2S production is phytoplankton emissions coupled to assimilatory sulfate reduction, not OCS hydrolysis, and most of the resulting H2S is complexed with trace metals such as zinc (Walsh et al., Limnol. Oceanogr.,39: 941-948, 1994; Radford-Knoery [NOTE CORRECT SPELLING] and Cutter, 1994; Cutter et al., Deep-Sea Res. II, 46: 991-1010, 1999).

We have rewritten the whole paragraph for clarity:

[revised manuscript text omitted]

---

## Author Comment (AC1)

**Reply to** Review of manuscript os-2019-1

"The influence of dissolved organic matter on the marine production of carbonyl sulfide (OCS) and carbon disulfide (CS2) in the Eastern Tropical South Pacific" by Lennartz et al.

**We thank the referee for the thorough and detailed feedback that helped us clarify some aspects in our argumentation. Please find our replies in blue font below the corresponding comments (page and line numbers refer to the old manuscript, in order to be consistent with the comments of the reviewer).**

1. General comments

This manuscript presents a comprehensive set of results from a cruise into the Peruvian upwelling area. Observational data are of high quality and unique in that the report OCS and CS2 concentrations together with a wider range of complementary information such as dissolved organic sulfur (DOS), DOM optical characteristics and diapycnal diffusivities. This comprehensive data set is complemented by applications of 1D models of trace gas biogeochemisty and mixed layer dynamics. This approach can allow for near synoptic interpretation although there are of course limitations arising from our incomplete understanding of underlying production mechanisms. The manuscript is clearly appropriate for publication in Ocean Science, not least due the unique combination of comprehensive data and modeling applications. However, I do have some concerns about presentation, and about some aspects of data interpretation and modeling results.

Title: The term 'Eastern Tropical South Pacific' used in the title suggests a study area much larger than that covered in the actual cruise track. I therefore suggest replacing ETSP with 'Peruvian upwelling'.

**Corrected as suggested (also replaced ETSP throughout the manuscript).**

**OCS photoproduction:** Interestingly the manuscript reports that OCS photoproduction is well correlated with colored dissolved organic matter (CDOM) characteristics such as humic-like fluorescence emission but not with DOS. This leads authors to speculate that the sulfur needed for OCS production may come from its hydrolysis product H2S and not from organic fractions. However, there are several issues the authors neglect to discuss. Firstly, the absence of a correlation between sulfur from the DOS pool and OCS photo production per se does not exclude that DOS fractions are involved in OCS production. Reported DOS concentrations exceed OCS concentrations 1000-fold (see figure 2 in MS). Therefore, only a tiny DOS fraction is required as the source of OCS sulfur. This tiny fraction might not necessarily correlate with bulk DOS. Secondly, although laboratory experiments showed that both H2S and free sulfurbearing amino acids such as cysteine may act as OCS precursors, precursor concentrations in these experiments were unrealistically high (10 μM) (Pos et al., 1998). S-OCS may well be derived from DOM itself given that irradiations of both unaugmented seawater and solutions of reconstituted DOM produced OCS (Zepp and Andreae, 1994; Pos et al., 1998). Thirdly, regarding H2S, available data suggest concentration levels of free sulfide below 100 pM (see e.g. Andreae et al., 1991). Although this may be similar to OCS concentrations, it is unrealistic sulfide to OCS. Clearly formation of the thermodynamically stable end product sulfate would be favored here. And finally, although the authors state that sulfurbearing amino acids 'are rare' (p 17, lines 9-11), they are still likely to occur at levels high enough to sustain picomolar OCS

levels. I recommend that the authors rewrite the corresponding sections in the light of my comments above.

**We thank the referee for pointing these issues out. The significant correlation to FDOM C2 indicates a tight connection to optically active DOM. This correlation would be less strong, if a rare sulfur bearing part of the DOM pool would be the limiting factor, unless this precursor fraction of the DOS pool covaried strongly with the FDOM fraction. We added this as a possible scenario in the text.**

**For clarity, we rewrote the paragraph in the Discussion:**

**p. 15, l.10: An interesting finding is the significant correlation of the photoproduction rate constant $p$ with FDOM C2 (humic-like FDOM), but not with $DOS_{SPE}$, given a reported correlation of OCS and DOS in the Sargasso Sea where much higher DOS concentrations of ca. 0.4 µmol S $L^{-1}$ were present (Cutter et al., 2004). It should be noted that the method to extract $DOS_{SPE}$ in our study does not recover all DOS compounds, and we cannot exclude the possibility that this influences the missing correlation between $p$ and DOS. In the studied area, OCS photoproduction is apparently not limited by the bulk organic sulfur, but rather by humic substances. The humic-like FDOM component C2 is an abundant fluorophore in marine (Catalá et al., 2015; Jørgensen et al., 2011), coastal (Cawley et al., 2012) and freshwater (Osburn et al., 2011) environments. This FDOM component seems to be especially abundant in the deep ocean (Catalá et al., 2015), which might be the reason for higher C2 surface concentrations in regions of upwelling, as evident in our study (Fig. 2) and reported by Jørgensen et al. (2011). The significant correlation of $p$ with humic-like fluorophores in our study highlights the importance of upwelling and coastal regions for OCS photoproduction.**

**A significant correlation (i.e., a limitation) of OCS photoproduction with humic-like substances, but not with bulk $DOS_{SPE}$ can be explained by two scenarios: Under the assumption that only organic sulfur is used to form OCS, the limiting factor is contained in the humic-like C2 fraction of the FDOM pool. The sulfur demand (75.8 pmol $L^{-1}$, the orange area in Fig. 7b) would need to be covered entirely by organic, sulfur-containing precursors. The limiting driver of this process is either organic molecules acting as photosensitizers or a sulfur-containing fraction of the DOM pool that correlates with FDOM C2, but not bulk $DOS_{SPE}$. In that scenario, FDOM C2 can be used as a proxy for the OCS photoproduction rate constant. More data from other regions would help to quantify such a relationship. In a second possible scenario under the assumption that both organic and inorganic sulfur can act as a precursor, the sulfur demand could theoretically be covered by the sulfur generated by hydrolysis of OCS (i.e. 85.8 pmol $L^{-1}$, Fig. 7). In this case, FDOM C2 would only be limiting as long as enough organic or inorganic sulfur is present, for example when temperatures are high enough to recycle sulfur directly from OCS, or when other inorganic sulfur sources are present.**

**Incubation experiments have shown that inorganic sulfur is a precursor for OCS (Pos et al., 1998). It is not clear whether the mechanism proposed therein occurs under environmental conditions, because sulfide concentrations were higher than in most marine areas, but also yielded much higher OCS production rates in the magnitude of nM $hr^{-1}$ compared to the magnitude of pM $hr^{-1}$ under natural conditions. Furthermore, the conversion of sulfide to sulfate, rather than to OCS, is thermodynamically favored. Based on our data, we cannot resolve the question about the role of inorganic sulfur in OCS photoproduction, but our results are consistent with the reaction**

mechanism suggested by Pos et al. (1998). Incubation experiments at environmentally relevant sulfide concentrations, as well as p-DOS relationships across different temperature and DOM regimes will help to resolve this issue.

We also changed the following in the conclusion:

p. 19, l. 26: The absence of a correlation of the photoproduction rate constant with bulk $DOS_{SPE}$ is not conclusively answered. Possible scenarios involve either a sulfur-containing precursor in a constant ratio FDOM C2, or an excess of sulfur via allocation from OCS hydrolysis. In both cases, FDOM C2 is a promising proxy on regional scales, and in case of scenario 1 also on global scales. ~~A possible explanation for the lack of correlation between the photoproduction rate constant and DOSSPE might be that OCS hydrolysis generates enough sulfide to recycle back to OCS during photoproduction. This cycle might be especially active in the ETSP, where warm temperatures increase the hydrolysis rate and, thus, the generation of sulfide, and abundant humic-like molecules further enhance 30 photoproduction.~~

We deleted the corresponding sentence in the abstract:

**OCS dark production:** The manuscript presents OCS dark production rates derived using a steady state assumption (methods, p 8). I agree that a steady state assumption may be made for samples "below the euphotic zone" or better for samples collected from below the MLD because sea surface OCS shows a pronounced diel cycle. However, I am not entirely convinced that "early morning" samples always reflect steady state. Half lives of OCS with respect to hydrolysis removal range from > 80 hours at 5ºC to ~3 hours at 30ºC (Elliott et al., 1989). For most of the cruise track half-lives were likely in the order of 10 hours. Given that OCS concentrations peak in the afternoon, and that early morning sampling will occur less than 2 half-lives later, OCS levels are likely biased i.e. more than 25% higher than 'steady state' assuming that hydrolysis is the main removal process. According to equation (2) artificially high OCS levels directly translate into artificially high dark production rates. Therefore, I would welcome some additional detail on sample selection, and some quantitative considerations of possible bias in section 2.8.

We changed and added the following:

p. 8, l. 7: Dark production rates were determined from hourly averaged measured seawater concentrations  shortly before sunrise (i.e. ca. 12-14 hours after concentration maximum of the previous day) or at depths below the euphotic zone.

p.8, l. 10: To ensure steady state conditions, we averaged the concentrations one hour before sunrise and compared to the average of the previous hour. We only considered instances when the concentration before sunrise deviated less than 1 pmol L$^{-1}$ from the previous hour for further calculation.

p.8, l.28: Biases can potentially be introduced in two ways: 1) neglecting other sinks like air-sea exchange can lead to underestimations of the production rate. With wind speeds of 8 m s$^{-1}$ and MLD on the order of 20-40m, life times due to air-sea exchange are in the order of days to weeks,

**and hence negligible. 2) Sampling less than two half lives after the maximum concentrations can lead to overestimations of the production rate. For the 11 and 12 October, samples considered for calculation of dark production rates were taken less than two half lives after the concentration maximum of the previous day. Since the concentration changed less than 1 pmol L$^{-1}$ within two hours prior to this sampling, we consider the bias as within the range of the given uncertainty.**

CS2 vertical profiles: On p 13 ff the authors present CS2 depth profiles which are then further discussed on p 17 line 22 ff together with modeled profiles. Unfortunately, the full set of profiles is only shown in the supplement, although these are clearly needed to support the results and discussion section. I therefore recommend moving figure S2 into the main paper.

**We include the figure in the main text and updated the figure numbering.**

I also have some issues with the modeling results shown both in figures 6 and S2. All simulations in S2 and most in Fig 6 show MLD concentrations below those in underlying waters. To me this seems to indicate that photoproduction in the MLD (the only source term used here) is lower than air sea gas exchange losses. However, if photoproduction is the only source term, why do these profiles indicate increasing concentrations with increasing depth across the thermocline? The modeled profiles in S2 all suggest a CS2 flux from below the TC into the mixed layer, although photoproduction should clearly be constrained to the MLD. Do these profiles show a non-steady state solution? Is it possible that the 1D model was initiated with a homogenous CS2 depth distribution rather than with CS2 free water? I think here the manuscript needs to give a much more detailed explanation of the modeling setup and a more cogent explanation of the resulting profiles.

**For clarification, we added the following:**

**p. 10, l.9: Profiles were initialized with the lowest subsurface concentration of the respective measured profile: low enough to be able to assess whether in-situ photoproduction can explain concentration peaks below the mixed layer, but high enough to keep diapycnal fluxes out of the mixed layer in a reasonable range (in contrast to initializing with 0 pmol L$^{-1}$). The same conditions that occurred on the day of measurement were repeated for 21 days, i.e. ~2-3 times longer than the lifetime due to air-sea exchange.**

**As stated, we only assess the modelled shape of the profile, not the actual magnitude. A longer simulation where photoproduction occurs below the mixed layer would then accumulate even more CS$_2$.**

I am also not entirely convinced of the authors' idea that 'deep' photoproduction below the MLD may have caused the observed subsurface maxima at stations 5 and 18. The statement on p 17 lines 34-5 "substantial production takes place at higher wavelengths penetrating deeper into the water column" needs to be put into context: based on apparent quantum yields used (Xie et al., 1998), light levels and KD, what is the depth dependence of CS2 photoproduction? How much of the total occurs below the MLD?

**Please note that these sensitivity tests only refer to station 5, as described in Tab. 1. For clarification, we added:**

**p. 17, l.22 : More detailed simulations were performed for station 5, because at this station, the photic zone extended below the mixed layer.**

**The statement on p 17, l.34-5 strictly refers to the scenario named at the beginning of the sentence/the previous sentence, i.e. the wavelength-integrated approach and not the wavelength-resolved approach by Xie et al. For clarity, we rewrote:**

**p. 17, l. 32: In our simulation 'pfit', a wavelength-integrated approach was adopted (eq. 6). Photoproduction is calculated with the integrated irradiation (300-400nm) and one rate constant, representing a wavelength-integrated AQY. In this simulation, photoproduction occurring at higher wavelengths, that are penetrating deeper into the water column, is higher compared to the wavelength-resolved simulation.**

**We include a new figure in the supplement, showing the photoproduction of CS2 in different model runs:**

[Figure]

**p. 14, l. 8: Photoproduction rates for these simulations are shown in S-Fig. 3 (see supplementary material).**

I wonder how this likely very small source term would compare to diapycnal transfer, and I very much doubt that it could sustain the elevated CS2 levels below the MLD.

**The point that we want to make here is that even a small source term can potentially accumulate, because the production occurs without the influence from air-sea gas exchange below the pycnocline (see p. 17, l. 34). Please note that the model already accounts for the diapycnal flux.**

Of course, the limited understanding of CS2 cycling hinders the modeling here. However, the treatment of CS2 depth profiles is rather unbalanced, with some data shown only in the supplement and not fully discussed, while Figure 6 only shows results for one station. The authors might want to consider rebalancing the reporting of their CS2 results.

**The seemingly unbalanced reporting results from 2 reasons:**

1) **Only four stations had additional parameters such as FDOM and $DOS_{SPE}$ to compare with concentrations and, more importantly,**
2) **Station 5 was the only station where the photic zone extended below the mixed layer and was thus chosen to assess photoproduction without the sink of air-sea exchange, since the mixed layer acts as a barrier.**

**Also in response to another comment below, we have changed:**

**p. 10, l. 17:**

**For the second test, demonstrating the sensitivity of the subsurface peak, we chose station 5. This station provides the unique opportunity to assess a profile where the photic zone reaches below the ML, hence photoproduction might occur at depths where the sink of air-sea exchange is absent due to the bottom of the mixed layer acting as a barrier.**

**In the discussion, p. 18, l. 13:**

**$CS_2$ was still detectable below 200 m, in concentrations around 5-10 pmol $L^{-1}$ in shelf regions and around 20 pmol $L^{-1}$ in open ocean regions (except station 1). This pattern reflects the spatial variation of surface concentrations, which were higher at the open ocean than at the shelf. The vertically relatively uniform concentration profiles suggest low degradation rates, and the travel distance of the water between the stations is too short to explain the concentration difference only by *in-situ* degradation. A Lagrangian approach would be helpful to resolve this issue.**

**Some profiles display small local maxima in the region of the oxycline (not shown), but due to unconstrained subsurface source and sink processes, no conclusion can be drawn on whether a chemical or a physical process is responsible. The rather homogeneous concentrations below 200 m depth suggest slow *in-situ* degradation rates. As a result, physical processes resulting from currents, eddies or shelf processes might gain a higher importance for the distribution of $CS_2$ in the subsurface compared to the shorter lived gas OCS. With sinks potentially acting on long timescales, $CS_2$ could possibly be transported from sources located further away, e.g. from contact to the sediment in shelf regions or subducted from the surface. Our results clearly show the limits of interpreting 1D concentration profiles for long lived compounds with both subsurface sinks and source unconstrained. Incubation experiments using isotopically labelled $CS_2$ would be helpful to constrain source and sink processes independently.**

2. Specific and editorial comments

Abstract:

The abstract is rather vague, does not give details on time period, sample numbers, and areal extent of the study area, and avoids any quantitative statements. Please rewrite.

**We changed the abstract to:**

**Oceanic emissions of the climate relevant trace gases carbonyl sulfide (OCS) and carbon disulfide (CS$_2$) are a major source to their atmospheric budget. Their current and future emission estimates are still uncertain due to incomplete process understanding and, therefore, inexact quantification across different biogeochemical regimes. Here we present the first concurrent measurements of both gases together with related fractions of the dissolved organic matter (DOM) pool, i.e. solid-phase extractable dissolved organic sulfur (DOS$_{SPE}$, n=24, 0.16±0.04 µmol L$^{-1}$), chromophoric (CDOM, n=76, 0.152±0.03) and fluorescent dissolved organic matter (FDOM, n=35) from the Peruvian upwelling region (Guayaquil, Ecuador to Antofagasta, Chile, October 2015). OCS was measured continuously with an equilibrator connected to an off-axis integrated cavity output spectrometer at the surface (29.8±19.8 pmol L$^{-1}$) and at four profiles ranging down to 136 m. CS$_2$ was measured at the surface (n=143, 17.8±9.0 pmol L$^{-1}$) and below, ranging down to 1000 m (24 profiles). These observations were used to estimate *in-situ* production rates and identify their drivers. We find different limiting factors of marine photoproduction: while OCS production is limited by the humic-like DOM fraction that can act as a photosensitizer, high CS$_2$ production coincides with high DOS$_{SPE}$ concentration. Quantifying OCS photoproduction using a specific humic-like FDOM component as proxy, together with an updated parameterization for dark production, improves agreement with observations in a 1D biogeochemical model. Our results will help to better predict oceanic concentrations and emissions of both gases on regional and, potentially, global scales.**

Introduction

Page 2 lines 26-7: "[CDOM…] and contains the photosensitizers that absorb light and form radicals for photochemical reactions". This statement should be revised: not all reactive species formed during irradiation of natural water DOM are 'radicals' (e.g. singlet oxygen, triplet states, solvated electrons) nor does CDOM act solely as a photosensitizer given that it undergoes photodegradation itself.

**We revised the statement:**

**Chromophoric DOM (CDOM) is the fraction that absorbs light in the UV and visible range. CDOM contains photosensitizers that absorb light and facilitate photochemical reactions, and can undergo photodegradation itself (Coble, 2007).**

Page 2 lines 31-2: "The method favors the retention of polar molecules, which comprise approximately 40 % of the total dissolved organic carbon (DOC) in marine waters". Please add supporting reference.

**We added the following reference: Dittmar et al. (2008)**

Page 3 lines 5 ff: "the CDOM absorption 5 coefficient at 350 nm (a350) can serve as a proxy for both photoexcitable carbonyl-groups and organic sulfur precursors making the overall photoproduction rate second-order dependent on a350 (von Hobe et al., 2003)". (i) a350 may be a proxy for DOM but not specifically for carbonyl groups and DOS. Simple carbonyls show absorption maxima well below

350 nm (e.g. aceton: ~265 nm). (ii) Relationships between OCS photoproduction and a350 were initially proposed by Uher and Andreae (1997) and subsequently applied to a global model by Preiswerk and Najjar (2000). This should be reflected here. (iii) It would be worth clarifying that 'second order' here should not be understood in terms of chemical kinetics. Instead, this statement refers to the dual roles of CDOM in light absorption and photochemistry.

**The reviewer is correct that the "second-order dependence of OCS photoproduction on the concentration of CDOM" stated in von Hobe et al. (2003) was not meant in the sense of an accurate kinetic rate law. Although based on known relationships with a solid physical and chemical reasoning, the p vs. aCDOM relationships in that paper were derived empirically (therefore, we consider the results semi-empirical), and we believe that they do at least broadly reflect the chemical kinetics at play. Also note that in that earlier paper aCDOM absorbance was explicitly measured at different wavelengths between 297 and 365 nm.**

**In the current paper, we have now**
1. **stated more clearly that $a_{350}$ here is used as a proxy for CDOM concentration and that both photosensitizers and organic sulfur compounds are expected to broadly correlate with CDOM concentration**
2. **Clarified that the relationship should not be understood as an exact chemical rate parameterization (stressing the word "proxy")**
3. **Mention the historical background with citations of Uher and Andreae (1997) as well as Preiswerk and Najjar (2000)**

 **p. 3, l. 5: Indeed, the amount of OCS produced has been shown to depend on CDOM, more specifically the absorption coefficient at 350 nm ($a_{350}$), and a variety of organic sulfur-containing precursors, such as methionine or gluthathione (Zepp and Andreae, 1994; Flöck et al., 1997). $a_{350}$ has been used as a proxy to calculate photochemical production of OCS previously (Preiswerk and Najjar, 2000). In addition, von Hobe et al. (2003) suggested a relationship between the photoproduction rate constant and $a_{350}$, making the overall photoproduction rate quadratic with respect to $a_{350}$. This dependency is based on the assumption that $a_{350}$ can serve as a proxy for both photosensitizers and organic sulfur precursors on large spatial scales.**

Page 3 lines 20 ff: Preiswerk and Najjar (2000) should be added here.

**Added as suggested**

Page 3 lines 31 ff, CS2 lifetime: Please clarify if this 'CS2 sink' should be airsea gas exchange or an additional unknown process. Gas exchange would have a matching lifetime in the order of weeks.

**We changed to:**

**In addition to the known sink, namely air-sea exchange, hydrolysis and oxidation, Kettle (2000) proposed a sink with a lifetime on the order of weeks, to match observed concentrations with a surface box model.**

Methods

Page 4 lines 23 ff, OCS calibration: Please provide a quantitative comparison of calibrations before and after the cruise. Why is the stated LOD (180 pptv) 12 times higher than the stated precision? What did you mean by "NOAA scale"?

**The precision is given for the two minute-averages (this is stated more clearly in Lennartz et al., 2017) and was experimentally determined (running a standard for > 60 minutes). Because the instrument internally analyses 1-second spectra, the LOD is determined by the 1-second precision, which is close to 180 pptv.**

**NOAA scale is the gaseous OCS standard used as a reference for NOAA time series stations. The instrument used for our measurements has been independently checked against a NOAA reference standard (Lennartz et al., 2017; Montzka et al., 2007).**

Page 4 line 29: Does "ca. 35 m" refer to the height of the inlet above sea level? It would also be interesting to see atmospheric OCS mixing ratios reported somewhere.

**We added "above sea level" to this statement.**

**This paper focuses on the processes in the water column, emissions based on the concentration gradient between surface water and atmospheric mixing ratios were already reported previously. We added the following statement:**

**p. 4, l. 30: Resulting emissions are reported in Lennartz et al. (2017).**

Page 5 lines 26 ff: Replace "spectrophotometer" with "spectrofluorometer". I suspect your statement in line 29 refers to photomultiplier voltage? Please state this clearly. Explain why you list two conversion factors for conversion between QSU and Raman units. How did you apply the QSU scale to fluorophores that do not show significant overlap with quinine sulphate emission?

**Spectrofluorometer and photomultiplier were corrected as suggested.**

**The two conversion factors originate from two different calibrations, carried out before and after the change in photomultiplier voltage. The QSU scale is a result of the PARAFAC analysis and was applied as described in Murphy et al. (2013). We added an explanatory sentence to p. 6, l. 4:**

**FDOM concentrations are reported here in quinine sulfate units (QSU, Murphy et al. (2013)), the conversion factor between QSU and Raman units is 0.3540 and 0.4256, for each of the QS calibration (i.e. before and after the change in photomultiplier voltage), respectively.**

Page 6 line 30: "downwelling irradiance profiles were corrected for incident sunlight". This statement is worded incorrectly. I suspect you adopted the profile from station 6 after normalization to sea surface irradiance? Please clarify.

**The correction was applied to all profiles, and means the correction of the irradiance measured in the water column to changes at the surface, e.g. due to clouds. We have inserted a linebreak after discussion of profile No. 6 to clarify this, and changed "the downwelling irradiance profiles" to "all downwelling irradiance profiles" in the following sentence. We also inserted "…were corrected for incident sunlight (e.g. changing due to varying cloud cover) using simultaneously obtained…."**

Page 7 line 7: Reference to equation 1 is incorrect. Equation 1 refers to diapycnal flux, not to underwater irradiance. Please insert appropriate equation and update equation numbering.

**The unnecessary equation reference was deleted.**

Page 7 lines 23-25: Please add a supporting reference.

**The supporting references are the references given in the sentence starting "Details in the methodology…." after we describe the microstructure sonde. To clarify this, we changed the order of these two sentences.**

Page 8 lines 1-2, dark OCS production: Please spell out the units used in equations (2) and (5).

**Units for equation 2 were given in 8, l. 11-12 (i.e. in the sentence explaining the equation directly before the actual equation). We added the units for $P_D$ again for equation 8, as well as the units for the coefficients a and b.**

Page 10 lines 13-15: Given MLDs in the order of 30-40 m and piston velocities of maybe 1-2 m d-1 , CS2 lifetimes with respect to ventilation losses should be in the order of weeks not days. Please revise your statement.

**changed as suggested**

Page 10 lines 17-19: Explain why "air-sea gas exchange is absent" at station 5.

**For clarification, we added:**

**For the second test, demonstrating the sensitivity of the subsurface peak, we chose station 5. This station provides the unique opportunity to assess a profile where the photic zone reaches below the ML, hence photoproduction might occur at depths where the sink of air-sea exchange is absent due to the bottom of the mixed layer acting as a barrier.**

Results

Page 10 lines 28ff, description of results in figure 2: Your statement that "DOM showed strong spatial variability in FDOM, but less in the DOSSPE concentration and CDOM absorbance" is not supported by data in figure 2. All three variables vary roughly 3-fold during your cruise, including FDOM component 2, particularly if highest & lowest FI were excluded from analysis. Unfortunately, however, mean±stdev are not reported for component 2 (nor for any other components). You stated that DOS decreased with depth but did not quantify this. Please rewrite this section and include the required statistical and quantitative information.

**We have now included the mean, standard deviation and coefficient of variation for all FDOM compounds. Given that FDOM coefficients of variations are in the range of 0.74 to 1.74, and those of DOSSPE and a350 0.31 and 0.2, respectively, we consider the statement concerning the spatial variation as correct.**

**We also added a quantification of the subsurface concentrations of DOSSPE, p. 11, l. 10:**

**Concentrations decreased from 0.76 (5 m depth) to 0.33 µmol L$^{-1}$ in 100 m at station 7, from 0.62 (25 m ) to 0.49 µmol L$^{-1}$ (125 m) at station 7 and from 0.49 (20m) to 0.28 µmol L$^{-1}$ (115m) at station 18. At station 2, concentrations of 0.89-0.91 µmol L$^{-1}$ were measured at a depth of 50-100m; no surface data is available.**

Page 11, OCS distribution: The authors should clearly state here and in their introduction that OCS sea surface concentrations from this cruise were already published in Lennartz et al (2017).

**We included a statement in the introduction and the discussion:**

**p.4 l. 1: Surface concentrations and emissions to the atmosphere from the cruise presented here are discussed in (Lennartz et al., 2017). Here, we focus on processes in the water column.**

**p. 11 l. 15: Surface concentrations as well as emissions to the atmosphere are described in detail in (Lennartz et al., 2017).**

Page 12, OCS photoproduction, lines 12 ff: It is interesting that OCS production rates only covary with humic-like fluorescence but not with a350. Please give further details of your analysis: how did you bin C2 and a350 data into days? Did you only use samples obtained between sunrise & sunset? Figure 6 only shows data for 6 days. How many days were available to test for pOCS – a350 correlations? Given that your experiment was not Lagrangian, how could changes in CDOM characteristics during diurnal cycles have affected your relationships? I would also be interested to see if a350 and C2 fluorescence were correlated with each other as they usually

**We have added the missing statistics on DOM covariation:**

**p. 12, l.15: Measurements of FDOM (and $a_{350}$) during the period used for optimization of the photoproduction rate constant $p$ (i.e. daylight period) were averaged for this correlation.**

**p. 12, l.20: The correlation with $a_{350}$ only explains a variance of R²=0.01 (n=7, i.e. 7, 8, 9, 10, 12, 13, 16 October 2015). R² increases to 0.3, when the respective days for FDOM C2 correlations are considered (p>0.25). C2 and $a_{350}$ were not significantly correlated during these days (p>0.2, R²=0.36), but showed a similar spatial trend all over the cruise track (Fig. 2). Although our experiment was not strictly Lagrangian, $a_{350}$ only changed <0.05 m$^{-1}$ within each respective fitting period. For FDOM C2, only 1-2 measurements per daylight period were available during the days when photoproduction rate constants were fitted, but variations of only 0.003 QSU per day were encountered during high frequency sampling towards the end of the cruise. This relationship thus carries some uncertainties, and will benefit from additional data from other regions.**

**Discussion**

Page 14, Carbonyl sulfide, lines 11-12: "[profiles] do not indicate any connection to a significant redox-sensitive process". This should be expected given that OCS photoproduction was found to be independent of dissolved oxygen concentration (Zepp and Andreae, 1994; Uher and Andreae, 1997). Please refer to previous work.

**We added:**

**p. 14, l. 13: The independence from dissolved oxygen concentrations is in line with previous findings by (Zepp and Andreae, 1994; Uher and Andreae, 1997).**

Page 14, line 17: Please remove reference to Ulshöfer et al. (1995) because they did not report dark production rates.

**Deleted as suggested**

Page 14, "radical production pathway", lines 25 ff: "A strong similarity across different biogeochemical regimes favors the hypothesis of a radical production pathway, which would be indifferent to the prevailing biological community". This statement is simply wrong and should be removed together with other associated statements. Available evidence clearly shows that rate constants of reactive radical species with DOM can vary significantly as a function of DOM source / composition (see e.g. McKay et al., 2011). Furthermore, DOM and its colored fraction are indirectly derived from primary production and therefore reflect its distribution pattern (see e.g. Carder et al., 1989), although not necessarily short terms variations. This unsupported speculation should be replaced by appropriate references to previous work. Kamyshny et al. (2003) for example, proposed a mechanism for dark production of OCS. That aside, possible differences in dark production rates between stations inside & outside the OMZ more likely reflect DOM compositional differences related to the microbial food web.

**Because of the rate constant varying as a function of DOM source, as the reviewer correctly states, we already normalized the rate constant of dark production by $a_{350}$ (see methods, e.g. equation 5). Normalizing by other parameters than $a_{350}$ does not improve the explained variation of the Arrhenius-relationship, as we describe in p. 14, l. 26ff. Our point is that abiotic parameters (i.e. $a_{350}$ and temperature), are better predictors for the dark production rate across very different biogeochemical regimes than biotic parameters. Of course, the microbial food web has a high impact on the local DOM composition, but this seems to be an indirect effect for OCS dark production, since it is best described using abiotic parameters.**

**To clarify this point, we changed:**

**p. 14, l. 26:**  **The fit in the Arrhenius-dependency could not be improved by other parameters than $a_{350}$, and showed no influence to dissolved $O_2$. The characteristics that make a molecule part of the CDOM pool, i.e. unsaturated bonds and non-bonding orbitals, also favor radical formation. OCS dark production is thus best described using abiotic parameters such as $a_{350}$ and temperature, than biologically sensitive parameters such as dissolved $O_2$ or apparent oxygen utilization as a proxy for remineralisation. This independence from biotic parameters supports the radical production pathway.(…)**

**The results are in line with findings by Pos et al. (1998) showing that these molecules can form radicals in the absence of light e.g. mediated by metal complexes, and by Kamyshny et al. (2003) showing a positive correlation of dark production rate and temperature.**

Page 15, line 5: "[…] internal waves led to high diffusivities". Explain how you arrived at this conclusion. What is the evidence?

**We have changed the sentence to:**

**There, high diffusivities were observed using the microstructure probe, which most likely result from high internal wave activity as indicated by vertical water displacements of up to 30m during four CTD casts.**

Page 15, correlations, lines 10 ff: see my previous comments on correlation results section 3.2.4.

**We have made the changes described above for clarification.**

Lines 18-20: give references supporting your statement regarding covariation of humic like fluorescence and a350.

**We changed:**

**On global scales, where p varies on a broader range than within the area covered by this study, $a_{350}$ might still be an adequate, but not perfect predictor for this variation (Lennartz et al., 2017). On local scales,  the parameterization for p based on a350 can be improved using FDOM C2.**

Page 15, lines 21 ff: see my previous comments regarding H2S as a possible substrate or OCS production.

**We made the changes described above in the general comments section to clarify our point.**

Page 16, lines 7 ff: This discussion of 1D model results refers to 'scaling factors' for AQYs. Note that AQYs cannot be assumed to be 'constant'. There is now clear evidence that AQY can vary with CDOM absorbance, presumably due to changes in DOM composition reflected by optical properties. See for example Gali et al. (2016) and Stubbins et al. (2011). The authors' own pOCS-CDOM relationships do reflect this as well. Establishing AQY-CDOM relationships for OCS would be useful and should be discussed in this section.

**This is exactly our point, so we clarified this further by adding/changing:**

**Our simulation X98 at stations 2, 5, 7 and 18 underestimates mixed layer $CS_2$ concentrations, indicating  spatial variations of the AQY, most likely due to changes in the DOM composition, as previously found for other gases (OCS: see above, carbon monoxide (Stubbins et al., 2011), DMS (Galí et al., 2016)). These results corroborate findings by Kettle (2000) and Kettle et al. (2001), who showed that the photoproduction of $CS_2$ was underestimated in some regions by the AQY from Xie et al. (1998). The scaling factor was on the order of 1-10 in Kettle's studies, which is in line with our results (factor 2-4). In future model approaches, a  photoproduction rate constant (expressing an integrated AQY) would need to be parameterized, and our results suggest that such parameterizations may rely on DOS or, on a global scale, DOC (since DOS covaries globally with DOC).**

Page 16, CS2 vs SST, lines 30 ff: Report the CS2 vs SST relationship in your data and compare this to Xie et al (1999).

**We added to the Result section:**

**p. 13, l. 6: Surface temperatures T [°C] and concentrations of CS$_2$ [pmol L$^{-1}$] were binned for daily averages, and yielded the following relationship (p=0.0026, R²=0.61) of (eq. 10):**

$$[CS_2] = 2.3\,T - 27.2 \qquad\qquad (10)$$

**And added to the Discussion:**

**p 16, l. 26: The significant relationship between surface temperature and CS$_2$ concentrations corroborates previous findings. Xie et al. (1999) found a positive correlation between CS$_2$ concentration and SST for the Pacific and the North Atlantic with a linear relationship of [CS$_2$] = 0.39t +7.2 (t=temperature in °C). Daily averages of our data close to the shelf (n=8, from 12 Oct onwards) between 15 and 20 °C fall within this relationship. However, daily averaged concentrations were higher than predicted according to this relationship further away from the coast at the beginning of our cruise at temperatures between 20 and 30°C (n=4). Overall, we confirm that CS$_2$ concentrations increase with increasing temperatures, but the exact relationship varies spatially. Reasons for this relationship could result from e.g. temperature-driven decay of precursor molecules, but remain speculative. The results are in line with findings by Gharehveran and Shah (2018), who found increased CS$_2$ formation with increasing temperatures in incubation experiments.**

Pages 17-18, CS2 photoproduction: see my previous comments regarding CS2 production and vertical profiles.

**We made the changes described above in the general comments section to clarify our point.**

Page 18, lines 7-9: Sustained subsurface maxima cannot be caused by 'downward mixing' because the CS2 gradient would cause transfer into the surface layer. Please remove this statement.

**We replaced "downward mixing" by "physical downward transport", since we are not referring to diapycnal mixing depending on the concentration gradient here (note that this process is already accounted for by the 1D model).**

**We added:**

**p.18, l. 9: Slow sinks below the ML would conserve high concentrations from surface waters due to the absence of air-sea exchange in the subsurface.**

Editorial:

The wording could be improved by careful editing.

[revised manuscript text omitted]

---

## Author Response (AR2)

**Reply to**

Comments to the Author:

Thank you for making the changes and additions suggested by the reviewers. I have only one additional request. In answer to one of the reviewers, you explained why the limit of detection for OCS is 12 x higher than the precision; please add this explanation to section 2.2. The paper can then be published.

**We thank Dr. Chapman for pointing this out, and adapt the manuscript by adding:**

**p. 4, l. 29 (new manuscript): The precision of this set-up is 15 ppt for two-minute averages of 1 Hz measurement frequency and was experimentally determined (running a standard for >60 min). The limit of detection is 180 ppt (corresponding to 4 pmol L$^{-1}$ at 20°C), defined by the instrument's internal 1-second spectra.**

**We also added a reference for the NOAA scale in the following sentence:**

**p. 5, l. 1: Additionally, independent samples for comparison measured with GC-MS (Schauffler et al., 1998; de Gouw et al., 2009) reflected <2% difference between the NOAA scale (Montzka et al., 2007) and the perm tube standards.**